# Novel Diagnostic Biomarkers in Colorectal Cancer

**DOI:** 10.3390/ijms23020852

**Published:** 2022-01-13

**Authors:** Aneta L. Zygulska, Piotr Pierzchalski

**Affiliations:** 1Department of Oncology, University Hospital, 2nd Jakubowski St., 30-688 Krakow, Poland; 2Department of Medical Physiology, Faculty of Health Science, Jagiellonian University Medical College, Michałowskiego 12 St., 31-126 Krakow, Poland; piotr.pierzchalski@uj.edu.pl

**Keywords:** diagnostic biomarkers, colorectal cancer, early detection

## Abstract

Colorectal cancer (CRC) is still a leading cause of cancer death worldwide. Less than half of cases are diagnosed when the cancer is locally advanced. CRC is a heterogenous disease associated with a number of genetic or somatic mutations. Diagnostic markers are used for risk stratification and early detection, which might prolong overall survival. Nowadays, the widespread use of semi-invasive endoscopic methods and feacal blood tests characterised by suboptimal accuracy of diagnostic results has led to the detection of cases at later stages. New molecular noninvasive tests based on the detection of CRC alterations seem to be more sensitive and specific then the current methods. Therefore, research aiming at identifying molecular markers, such as DNA, RNA and proteins, would improve survival rates and contribute to the development of personalized medicine. The identification of “ideal” diagnostic biomarkers, having high sensitivity and specificity, being safe, cheap and easy to measure, remains a challenge. The purpose of this review is to discuss recent advances in novel diagnostic biomarkers for tumor tissue, blood and stool samples in CRC patients.

## 1. Introduction

Colorectal cancer (CRC) is one of the most prevalent and incident cancers worldwide, as well as a significant cause of mortality. If colorectal cancer is detected in early stage, it is curable. Therefore, early detection can reduce mortality of colorectal cancer.

Colonoscopy has been well established as the gold standard of colorectal cancer screening with high sensitivity and specificity. However, it is costly in terms of money and manpower, requiring experienced endoscopists and patient adherence. The development of advanced molecular techniques comes with aid in detection and treatment of colorectal cancer. Currently, proteins, DNA (detection of mutations and methylation markers), RNA (mainly microRNAs), volatile organic compounds, changes and shifts in gut microbiome composition are the categories of colorectal biomarkers that have been explored [1,2].

Depending on the biological origin of material or the type of tissue being tested, we can distinguish biomarkers from the blood and stool. 

Nowadays, colorectal mucus, urine, saliva and exhaled air are sources of additional diagnostic options/biomarkers [1].

The purpose of this review is to discuss recent advances in novel diagnostic biomarkers for tumor tissue, blood and stool samples in CRC patients. 

## 2. Blood Biomarkers 

In cancer diagnostics, biomarkers might be different types of circulating biochemical molecules, such as proteins, tumor DNA, tumor-derived cells and miRNA in the blood, all of which are frequently used in CRC diagnostics. 

These biomarkers are detected in blood-based protein quantification tests or with immunohistochemistry [3,4]. Locally advanced malignant lesions enhance the level of circulating nucleic acids by up to 15-fold, the concentration in patients with metastatic cancers reaching up to 500 ng/mL [3]. The ease with which blood can be collected or donated means that the detection of blood-based biomarkers could be a practical screening tool for CRC. 

Genetic anomalies occur in a large fraction of sporadic non-hereditary cancers in the early stages of tumorogenesis. Large numbers of cells carrying these anomalies are shed from the developing tumor and can be found in biological effluents, mainly in stools, serum and urine as cell-free nucleid acids. The genetic anomalies can be easily identified by quick, non-invasive, relatively inexpensive molecular biomarkers with higher sensitivity and specificity than the feacal occult blood test (FOBT) or faecal immunochemical test (FIT) [5].

### 2.1. Liquid Biopsy 

Liquid biopsy detects circulating tumor cells from any bodily fluid, including peripheral blood, urine and cerebrospinal fluid, ascites, pleural effusion, etc., and includes a genomic or proteomic assessment. It is a quick, easy and low-budget method with minimal invasiveness, making it widely accepted by patients, without major side effects [5,6,7,8,9,10,11,12]. 

Liquid biopsy (using mainly peripheral blood) can be used as a screening method to detect early-stage CRC and minimal residual disease after surgery, or identify the molecular profile of CRC, the risk of recurrence, therapeutic targets, mechanisms of drug resistance and can influence the course of therapy in early CRC patients. In addition, liquid biopsy testing might allow the assessment of prognostic and predictive biomarkers in metastatic CRC patients [8,9,10,11]. However, currently, the clinical utility of liquid biopsies in CRC is limited. There is a need for better standardization and validation of liquid biopsy assessment [10]. 

Recent studies using liquid biopsy biomarkers in screening/diagnostics of CRC are collected in Table 1. 

### 2.2. Circulating Tumor Cells (CTC)

CTCs are epithelial cancer cells from the primary tumor or metastases which gain access to the circulatory system and are detectable in peripheral blood [6,11]. They could be used as biomarkers to detect CRC or to provide information about mechanisms concerning dissemination and enable decisions about therapy. Nowadays the widely used gold standard and still only FDA-approved method for CTC detection is a highly sensitive assay combining immunomagnetic enrichment with multiparameter flow cytometric and immunocytochemical analysis (the CellSearch System) [24]. However, CTC estimation has some limitations, e.g., blood concentration of CTCs is extremely low, which makes this method unreliable as a diagnostic tool [25]. The number of circulating CTCs ranges from 1–10 cells per 10 mL blood and can be even lower in early-stage cancers [26]. In order to enrich CTCs from whole blood, microfabricated trapping chambers are put in microfluidic chip devices. The CTCs are isolated based on differences in size and shape between tumour and normal blood cells [27]. The results require further validation studies. The prognostic value of CTCs in CRC has been proven but their use in screening still remains controversial [9]. On the other hand, Tsai et al. reported an overall accuracy of 88% for all CRC stages, including precancerous lesions using a new CTC assay [13]. 

### 2.3. Circulating Tumor DNA (ctDNA)

While cell-free DNA (cfDNA) was first reported in the nineteen-forties, when non-cell-bound nucleic acids were detected in the blood of cancer patients, the use of cfDNA as a diagnostic biomarker in daily practice started in the twenty-first century [28]. In many studies, high levels of cfDNA have been described in cancer patients. cfDNA fragments measure between 180 and 200 base pairs and mainly derive from apoptotic or necrotic tumor cells [29,30,31,32]. They can also originate from living tumor cells and circulating tumor cells. 

Caspases are cysteine proteases belonging to a large family which includes serine, aspartic acid and metalloproteases. Proteases may induce the progression of neoplastic disease as they degrade the extracellular matrix and contribute to the invasion of tumor cells into adjacent normal tissue [33]. The deregulation of caspases leads to the release of DNA or nucleosomes into the blood circulation [34,35]. 

cfDNA contains tumor-specific genomic alterations, such as mutations, methylation, loss of heterozygosity and microsatellite instabilities [36,37]. Their half-life in the circulation ranges from 16 min to 2.5 h [38]. Indeed, the overall quantity of cfDNA is 24-fold higher in the serum than in plasma, but plasma samples are preferable for ctDNA analyses [39,40]. 

Besides neoplastic processes, cfDNA concentrations are increased in such diseases as acute brain injury, acute ischemic stroke, infection and even after organ transplantation or incremental exercise [41,42,43,44,45]. 

Genotyping ctDNA in the blood to determine a tumor profile seems to be a good diagnostic approach. The ctDNA assay is a quick, easy, minimally invasive method, enabling a comprehensive tissue profile. However, limited evidence for treatment selection in advanced cancer and low correlation with histology or cellular phenotype are the main disadvantages of this method [46]. 

Nowadays, there are many conventional and nanomaterial-based methods for the detection of ctDNA, such as next-generation sequencing (NGS), the amplification refractory mutation system (ARMS), peptide nucleic acid clamping polymerase chain reaction (PNA-PCR), droplet digital polymerase chain reaction (ddPCR), PCR products analyzed by surface-enhanced Raman spectroscopy (SERS), electrochemical DNA biosensors (the means of detection based on generated electronic signals), peptide nucleic acid clamping asymmetric polymerase chain reaction and liquidchip (PAPL), ddPCR analysis based on microfluidic devices (IC3D ddPCR), BEAMing (beads, emulsion, amplification, magnetics) and Fe–Au nanoparticle-coupling strategy (a means of hybridization-based detection) [47]. 

ctDNA has been widely evaluated as a novel biomarker for liquid biopsy in colorectal cancer diagnosis, prognosis and monitoring of response to treatment. Liquid biopsy based on ctDNA detection is a very sensitive test. Bettegowda et al. demonstrated that the sensitivity of ctDNA for revealing clinically relevant KRAS gene mutations was 87.2% and its specificity was 99.2% in a group of 206 metastatic colorectal cancer patients [48]. 

### 2.4. Circulating MicroRNA (c-miRNA)

MicroRNA, a subclass of non-coding RNAs, was first discovered and characterised in the nineteen-nineties in *Caenorhabditis elegans* [49]. Interestingly, more than 38,500 human miRNAs have been described since then [50]. Moreover, circulating miRNA has played a role in diagnostics since the twenty-first century. 

MicroRNAs participate in proliferation, migration, differentiation, hematopoiesis, cell cycle regulation and stemness [51]. Dysregulation of miRNA’s function is a trigger of numerous diseases, e.g., cancer. 

MicroRNAs arise from intergenic, exonic and intronic sequences from the DICER–dependent pathway in the nucleus [52]. 

Initially, RNA polymerase II (POLR2) or RNA polymerase III (POLR3) take part in miRNA formation [53,54]. Primary transcripts (pri-miRNAs) several kilobases in length, which are 5′ capped and 3′ polyadenylated, are performed in the first stage of biogenesis. Subsequently, pri-miRNA is transformed into pre-miRNA by the microprocessor complex consisting of ribonuclease (RNase) III enzyme DROSHA and its cofactor [55]. The pre-miRNA is transported to the cytoplasm via Exportin-5 with Ran-GTP. The pre-miRNA is digested with the RNase III enzyme DICER and mature miRNA is produced [56]. There are alternative methods of miRNA maturation, such as DICER-independent biogenesis, DROSHA/DGCR8-independent biogenesis, the endogenous siRNA strategy and small nucleolar RNA-derived miRNAs [57,58,59,60]. 

miRNAs interact with the RNA-induced silencing complex (RISC) to bind to the 3′ untranslated region of mRNA molecules and regulate transcription and mRNA stability [61,62]. 

Circulating miRNAs play a role in communication between cells and influence gene expression in distant and adjacent target cells [63]. miRNAs are involved in post-transcriptional regulation of more than 60% of all protein-coding genes [64].

Circulating miRNAs is transferred associated mainly to exosomes or forming complexes with proteins such as Ago2 (Argonaute) [6]. 

miRNA concentrations are higher in plasma than in serum. Only extremely up- or downregulated miRNAs seems to be suitable to serve as clinical biomarkers [65]. In comparison to normal mucosa, close to two-thirds of miRNAs were downregulated in CRC [66]. Bartley et al. described 230 miRNAs that were differentially expressed in the four-stage transformation from nonneoplastic mucosa of the large bowel through low-grade dysplasia and high-grade dysplasia in adenoma to colorectal adenocarcinoma [67]. 

Dysregulation of miRNAs occurs frequently in adenoma and CRC. Therefore, they are a potential source of optimal biomarkers. The most common dysregulations are collected in Table 2.

Microarrays are the technology that make it possible to test simultaneously thousands of genes for differential expression. This technique is used to estimate miRNA profile changes in colorectal adenomas and CRC. Storage conditions of samples are very important: miRNA levels are stable at room temperature for at least 24 h in whole blood samples, while miRNAs are degraded at room temperature in serum samples; therefore, serum miRNAs should be stored for up to 72 h, refrigerated at 4 °C or frozen at −20 °C, because of the reduction of RNase activity at low temperatures [81,82]. 

RT-qPCR, digital PCR (ddPCR), hybridization-based techniques (e.g., microarrays, NanoString) and next generation sequence (NGS) are methods used to profile miRNAs [83]. 

It is necessary to establish clinically valid diagnostic techniques (RNA extraction, reverse transcription, qPCR analysis) using these markers as well as their optimisation and standardisation [6,65]. 

Nowadays, single miRNA markers of CRC, as well as combinations of miRNA markers detectable in plasma or serum, were also tested as diagnostic panels. Unfortunately, most of these studies gave modest or inconsistent results due to the fact that single miRNA has limitations in terms of low sensitivity and specificity. The levels of single miRNA molecules in CRC patients and healthy subjects frequently overlapped. Furthermore, the method of sampling should be able to distinguish patients with cancer from healthy individuals [84]. 

It is necessary to control hemolysis in samples in a preanalytical phase because hemolysis may change levels of circulating miRNA in samples due to the rupture of erythrocytes containing miRNAs [85]. There is still a lack of consensus on an adequate endogenous normalizer to quantify circulating miRNA expression in CRC [6].

Optimal conditions for processing blood specimens for miRNA measurement need to be established (e.g., in order to avoid the contamination of platelets in samples) [86]. Moreover, it is difficult to determine whether miRNA in the circulation is cancer-specific or co-morbidities specific [87]. 

MicroRNA variants investigated as potential diagnostic CRC markers are listed in Table 3.

### 2.5. Sept 9 Methylation 

Promoter CpG island methylation occurs more frequently than genetic mutation in cases of CRC. Hypermethylation contributes to carcinogenesis mostly by inducing transcriptional silencing or downregulation of tumour suppressor genes. Over 600 hypermethylated genes have been identified so far. Better understanding of epigenetics and technological progress enables the translation of laboratory research into daily clinical practice tools [114]. 

In humans, there are 13 genes (*SEPT1–SEPT13*) in the *SEPT* gene family. The *SEPT9* gene is located on the human chromosome 17q25 [115]. 

The best-known blood marker is SEPT9 methylated DNA. The detection rate of this method ranges from 57% to 64% in patients with CRC stages 0–I [7]. The combination of the detection of *SEPT9* promotor region methylation in plasma with FIT increased the sensitivity to 94% [103]. A meta-analysis published in 2017 including 25 research articles found that the *SEPT9* assay is only superior to the FIT in the symptomatic population [116]. The next meta-analysis confirmed that the rate of m*SEPT9* positivity was higher in advanced CRC cases compared with early-stage CRC cases and was higher in CRC than in adenomas. Diagnostic sensitivity was not correlated with cancer localization (left-sided versus right-sided) but with the race of tested individuals (being superior among Asian compared with Caucasian populations) [117]. The updated meta-analysis published in 2020 revealed that the *SEPT9* assay has a sensitivity of 69% and a specificity of 92% for CRC diagnostics. 

This test exhibited a poorer performance in detecting precancerous lesions (advanced adenomas and polyps) and is an expensive method [118]. 

In 2016 the FDA approved as a screening tool the Epi proColon Kit version 1.0 (which later was upgraded to version 2.0) that detects *SEPT9* promotor region methylation in plasma-originated tumour DNA, which is considered to be a specific biomarker of early CRC stages [119,120]. 

The *SEPT9* methylation assay is a qualitative test, which can be interpreted as positive or negative according to the application of different algorithms. Positive test results can be defined by one positive count out of three PCRs (1/3 algorithm), three positive counts out of three PCRs (3/3 algorithm), two positive counts out of three PCRs (2/3 algorithm) or one positive count out of one PCR (1/1 algorithm). Sensitivity increased with the decline in the required number of positive PCR reactions, while specificity increased with the rise in the required number of positive PCR reactions [121]. 

In the prospective PRESEPT (PRospective Evaluation of SEPT) study, the methylated *SEPT9* assay demonstrated a sensitivity of 48% for CRC (for stage I, 35%; II, 63%; III, 46%; and IV, 77%) and a specificity of 92%. Sensitivity was very low for advanced adenomas (14.4%) and only slightly higher than the false positive rate for all subjects who did not have cancer [122]. 

The US Preventive Services Task Force and the American Chemical Society currently do not include the Epi proColon test in their CRC screening guidelines due to its lack of effectiveness in detecting important precancerous lesions [123]. (An update of screening guidelines is currently in progress.) Recent studies using the *SEPT9* gene methylation assay in diagnostics of CRC are shown in Table 4.

### 2.6. Long Non Coding RNA (lncRNA)

Within the last decade a number of researchers have demonstrated that long non-coding RNAs (lncRNAs) are stable in blood and have diagnostic potential [148,149,150,151]. Based on length, ncRNAs are divided into two classes: small non-coding RNAs, whose length is a maximum of 200 nucleotides (i.e., microRNAs (miRNAs), small interfering RNAs (siRNAs), PIWI-interacting RNAs (piRNAs)) [152,153]; and long non-coding RNAs (lncRNAs), which have 200 or more nucleotides and which influence cancer cells through mechanisms such as chromatin remodeling, chromatin interaction, competing endogenous RNAs and natural antisense transcripts [154]. lncRNAs can be found in different body fluids, such as blood, plasma/serum or urine, due to the fact that it can cross cell membranes [155]. They derive from both living cells in an active manner and apoptotic cells. 

Many lncRNAs are associated with all stages of tumorigenesis and progression of CRC [154,156]. Their altered expression can influence several oncogenic signaling cascades, including the WNT/β-catenin, PI3K/Akt, EGFR, NOTCH, mTOR and TP53 signaling pathways [157,158,159]. Interestingly, circulating RNA directly represents the expression level of certain genes, which could distinguish cancer patients and healthy subjects [160]. Moreover, circulating RNA are characterised by their relative resistance to degradation by RNase enzymes [161,162]. 

Apoptotic bodies, microvesicles and exosomes are extracellular phospholipid-enclosed vesicles which travel with lncRNAs in the blood. These extracellular vesicles are released by cells into their environment. Their function is to transfer DNA, RNA, lipids, proteins, glycans and metabolites among cells [163,164,165]. There are three subgroups of extracellular vesicles. Apoptotic bodies 50–500 nm in diameter are released from cells undergoing programmed cell death. Microvesicles 50–1000 nm in diameter are produced by cell membranes [164,166]. Exosomes have intracellular origins and are 30–150 nm in diameter [167]. Among the three types of extracellular vesicles, exosomes express the highest levels of long micro RNAs (lmiRNAs) and contribute to chemoresistance, immunomodulation and metastasis in various tumor types [148,168]. It is estimated that around 2 × 10^15^ exosomes occur in the blood of healthy people. Exosome numbers can reach 4 × 10^15^ in case of cancer [169,170]. The most commonly used method for separating and accumulating vesicles is ultracentrifugation. This method is time-consuming and requires large amounts of starting material and expensive equipment. The commercial kits for exosome isolation are not standardized and the purity of results is low [171]. 

Quantitative reverse-transcription polymerase chain reaction (qRT-PCR) is the most used method of identification of circulating miRNA due to its high sensitivity and specificity [172]. The data obtained with qRT-PCR requires a normalization process in order to avoid potential technical bias which can be caused by pre-analytical or technical factors arising from miRNA extraction through to the amplification process. Unfortunately, there are no known adequate endogenous control or housekeeping genes to normalize the expression of miRNAs in different plasma or serum samples in cases of CRC. Duran-Sanchez et al. demonstrated that detection of miR-1228-3p showed no significant differences between hemolytic and non-hemolytic serum samples. Validated miR-1228-3p seems to be an adequate endogenous control for circulating miRNA analysis in CRC and advanced adenoma liquid biopsies [173]. 

CCAT1 and HOTAIR were the first markers reported to have significantly elevated expression in the plasma of CRC patients compared to healthy subjects [174]. Many other circulating lncRNAs have been described as potential biomarkers for CRC detection (e.g., LOC285194, RP11-462C24.1 and Nbla12061, 91H, PVT-1 and MEG3, NEAT1 variant 1 and NEAT1 variant 2); they are presented in Table 5.

Meta-analyses have attempted to summarize the potential clinical usefulness of circulating miRNAs in early CRC detection [92,175,180,181,182]. It seems that the simultaneous detection of entire panels of RNAs instead of a single circulating lncRNA increases the sensitivity and specificity of this diagnostic tool. 

### 2.7. Insulin-like Growth Factor Binding Protein 2 (IGFBP-2)

Various growth factors, e.g., insulin-like growth factors (IGF-1 and IGF-2), stimulate tumor growth and play a role as mitogens for colon mucosa [183,184,185,186]. IGFBP-2 is an extracellular protein binding insulin growth factor 2 (IGF-2) which participates in heat shock protein 27-mediated cancer progression and metastasis [187]. Both IGFBP-2 and IGFBP-1 participate in the regulation of cell proliferation, regulation of differentiation and apoptosis prevention. An increased level of IGFBP-2 in serum correlates with the presence of neoplastic alterations in the colon and concentrations of carcinoembryonic antigen (CEA) in colon cancer patients [188,189]. Therefore, assessment of IGFBP-2 levels has been suggested as a diagnostic parameter in the surveillance of patients with colorectal cancer. In turn, IGFBP-2 overabundance during colorectal carcinogenesis reduces tumor growth by the inhibition of cell proliferation [188]. 

### 2.8. Pyruvate Kinase M2 (PKM2)

PKM2 is an isoform of cytosolic enzyme pyruvate kinase (PK) involved in energy metabolism and is present in normal and cancer cells. Overexpression of PKM2 has been reported in CRC, gastric cancer and colon adenomas [190,191,192,193]. 

PKM2 seems a useful blood and faecal biomarker for CRC screening with its high sensitivity [191,193,194,195,196,197,198,199]. The disadvantage of this potential diagnostic biomarker is its poor specificity. 

### 2.9. Dickkopf3 (DDK3)

Tumor endothelial markers (TEMs) are a group of 46 genes showing stronger expression rates in the tumor endothelium of CRC tissues than in the endothelium of normal colonic mucosa [200]. The human cysteine-rich Dickkopf family includes Dkk-1, Dkk-2, Dkk-3, Dkk-4 and a unique Dkk-3-related protein called Soggy (Sgy), which belong to the class of TEMs [201]. 

The Dickkopf family are glycoproteins of 255–350 amino acids that contain a signal sequence of nuclear transport and share two conserved cysteine-rich domains, each of which displays a characteristic spacing of cysteines and other conserved amino acids. The N-terminal cysteine-rich domain—Dkk N (formerly termed Cys1)—is unique to the Dkks, while the C-terminal cysteine-rich domain–formerly termed Cys2—has a pattern of ten cysteines related to that described in the colipase family. These two domains are separated by a non-conserved linker region [202]. 

Activation of the Wnt signalling pathway participates in tumorigenesis, and the epigenetic silencing of Wnt antagonist genes, among others, is detected in CRC. Such antagonist genes are DDK genes which are epigenetically silenced in CRC cells due to promoter hypermethylation. Downregulation of Dkks expression by small interfering RNA (siRNA) contributes to the growth of cancer cells and invasiveness in vitro [203,204]. 

Dickkopf-3 (Dkk-3) is different from the human Dkks family, and its biological function is unclear. Dkk-3 possesses an extended N-terminal domain that precedes the Cys1 domain and an extended C-terminal region that follows Cys-2 [201]. It seems that Dkk-3 is not involved in inhibition of the Wnt signalling pathway. Zitt et al. reported that Dkk-3-positive CRCs revealed a statistically significantly higher mean microvessel count (9.70 vessels) than Dkk-3-negative cancers. Therefore, Dkk-3 is considered as a pro-angiogenic factor involved in neovascularization during CRC growth [205]. 

### 2.10. DDK3, PKM2, IGFBP-2

This biomarker model is able to identify early-stage CRC with 57% sensitivity for stage I and 76% sensitivity for stage II at 95% specificity. Therefore, this panel of biomarkers seems to be used as a non-invasive blood screening and/or diagnostic test and is equivalent to FOBT and FIT in terms of detection of CRC [206]. 

## 3. Tissue Biomarkers

### 3.1. Caudal Type Homebox 2 (CDX2)

Caudal type homeobox transcription factor 2 (CDX2) is a homeobox protein which is expressed in the nucleus of the intestinal epithelial cells. It plays a key role in the embryonic formation and differentiation of the intestine [207,208]. CDX2 is widely used as a sensitive and specific immunomarker for CRC. CDX2 is perceived as a tumor suppressor gene in CRC because its expression is lacking in CRC cases. Overexpression of CDX2 reduced colon cancer formation in mice. Downregulation of Ki-67 was associated with this process [209]. Overexpression of CDX2 using a hypoxia-inducible human telomerase reverse transcriptase (hTERT) promoter-driven vector suppressed malignant progression of colon cancer cells both in vivo and in vitro [209]. 

CDX2 expression is regulated by exchange proteins directly activated by cAMP (the Epac pathway) [208].

The Wnt/β-catenin pathway is an important signalling pathway in embryonic development and tissue homeostasis. Aberrant β-catenin expression and overactivation of Wnt signaling participates in CRC commencement [210]. CDX2 inhibits the transcriptional activity of β-catenin in a non-transcriptional way [211]. Toth et al. showed statistically significant correlations between sustained nuclear CDX2 expression and cytoplasmic β-catenin through Mucdhl in liver metastases of CRC [212]. There are two CpG-rich sites in the promoter region of the *CDX2* gene, −1570 to −1200 and −220 to +880. The upper CpG-rich site was heavily methylated in all cell lines. The lower CpG-rich site was methylated in limited CRC cell lines and was associated with down-regulation of CDX2 [207]. 

The methylation of the *CDX2* gene promoter region was associated with a high risk of CRC. The methylation rate of the promoter region of the CDX2 gene in the CRC tissue was 78.5% [213]. A lack of *CDX2* expression in metastatic CRC was significantly more frequent in women with right-sided primary mucinous tumors and poorly differentiated histologic features and distant lymph node metastasis [214,215]. 

There were positive correlations observed between CDX2 downregulation and both high-grade and more advanced tumors with liver metastases [212,214,215,216]. Moreover, low CDX2 expression combined with stage T4 CRC was associated with significantly lower disease-free survival and overall survival [217]. Rajarajan et al. noted that more than half the CDX-2 positive CRCs were without metastases to regional lymph nodes [218]. 

Nishiuchi et al. revealed that patients with CDX2-negative CRC in stage II/III had a five-year overall survival and relapse-free survival, lower than patients with CDX2-positive CRC. miR-9-5p regulates CDX2 expression. High miR-9-5p expression is positively correlated with a poor prognosis in patients with stage II/III CRC [219]. 

Dalebra et al. observed that patients with CDX2-negative colon cancers had a significantly shorter five-year disease-free survival (DFS) compared with patients with CDX2-positive cancers. Moreover, in individuals treated with adjuvant chemotherapy, CDX2-negative colon cancers in stage II were found to have a significantly increased five-year DFS (91%) in comparison with those who did not undergo adjuvant chemotherapy (56%) [220]. 

In another study, the authors showed that being CDX2-negative in the mesenchymal subgroup (CMS4) (of the consensus molecular subgroups classification of CRC) was associated with a poor prognosis for both overall and relapse-free survival [221]. The median overall survival for CDX2-negative patients was eight months and for CDX2-positive metastatic CRC patients was 39 months. The median of progression-free survival for first-line chemotherapy was significantly decreased in patients with CDX2-negative metastatic CRC (three versus ten months) [215].

Downregulation of CDX2 positively correlated with poor differentiation and mismatch repair (MMR) deficiency in CRC [212,222]. CDX2 loss was also associated with microsatellite instability [223]. 

Inflammatory bowel disease increases the risk for developing colitis-associated colorectal adenocarcinoma (CAC). Lack of CDX2 and YES-associated protein-1 (YAP1) expression occurred in younger patients and higher stages of CAC. Neither CDX2 nor YAP1 expression alone correlated with more aggressive histopathological features of CRC in patients with inflammatory bowel disease [224].

Interestingly, CDX2 loss is an independent poor prognostic marker in metastatic CRC patients [223]. Loss of CDX2 expression occurred more frequently in BRAF-mutated than in KRAS-mutated metastatic CRC (53% versus 9%). In other words, expression of CDX2 in BRAF-mutated cases was associated with a better prognosis and loss of CDX2 in KRAS-mutated cases was associated with a worse prognosis [223]. Patients with CDX2 loss more often had distant lymph node metastases than liver metastases. Immediate progression in first-line combination chemotherapy was seen more frequently in groups of patients with CDX2-negative than in CDX2-positive CRC tumors. Moreover, it was observed that patients with CDX2 loss had both lower overall survival and progression-free survival [223].

### 3.2. Special AT-Rich Sequence-Binding Protein 2 (SATB2)

Special AT-rich sequence-binding protein 2 (SATB2) is a part of the family of matrix attachment region-binding transcription factors regulating skeletogenesis (skeletal development and osteoblast differentiation) [225]. SATB2 upregulates osteoblast-specific genes. 

SATB2 represses the expression of Hoxa2 and activates several osteoblast-specific genes by enhancing the activity of the transcription factors Runx2 and ATF4, which regulate osteoblast differentiation. This shows that SATB2 plays a role in a transcriptional network regulating skeletogenesis. SATB2 is expressed in branchial arches and in cells of the osteoblast lineage. Satb2-/- mice exhibit craniofacial abnormalities, and the same abnormalities are observed in humans carrying a translocation in SATB2 or defects in osteoblast differentiation and function. This synergy was genetically confirmed by bone formation defects in Satb2/Runx2 and Satb2/Atf4 double heterozygous mice [225]. Initially, SABT2 was reported as a gene on 2q32–q33 involved in isolated cleft palate defects by FitzPatrick et al. [226]. In the following years, SABT2 has been recognized as a colorectal cancer biomarker and several genetic syndromes associated with SATB2 have been described [227,228]. A phenotype including intellectual disability, craniofacial abnormalities (e.g., cleft palate), dental malformations and osteopenia with SATB2 defects was reported by Zarate et al. [228]. Furthermore, SATB2 was identified as a potential immunohistochemical marker of human colorectal epithelium abnormalities through screening of the Human Protein Atlas database by Magnusson et al. [227]. The researchers characterized the expression profile of SATB2 in normal human tissues using tissue microarrays. SATB2 was highly expressed in the epithelium of the appendix, colon and rectum as well as in the cerebral cortex and hippocampus, non-germinal center lymphoid cells and the ductal epithelium of the testis and epididymis [227]. SATB2 has a relatively narrow expression profile mainly in colorectal/appendiceal adenocarcinomas [229]. Its specificity is 91.2% for metastatic gastrointestinal adenocarcinoma [230]. 

SATB2 is relevant to a number of potential applications, including determining the origin of adenocarcinomas of unknown primary and distinguishing primary ovarian mucinous adenocarcinomas from colorectal metastases [229,231]. According to Magnusson et al., using SATB2 as a solitary marker, SATB2 was positive in 83.7% of stage III/IV colorectal adenocarcinomas, 91.4% of stage II and 92.4% of stage I of this malignancy. Based on these results and the results of the tissue microarray from other common malignant lesions, researchers suggested that the combination of SATB2 with cytokeratin 20 (CK 20) is a highly specific marker for colorectal adenocarcinoma [227].

### 3.3. Glycoprotein A33 (GPA 33) 

The A33 antigen is a type I transmembrane glycoprotein of the immunoglobulin superfamily expressed in the basolateral membranes of both the proliferating cells in the lower regions of the crypts and the differentiating cells in the upper regions of crypts in the colon and in the small intestine, as well as in 95% of colon tumors [232,233]. It is a highly persistent, immobile, surface-localized protein [234]. GPA 33 is transcriptionally regulated by CDX1 in the tissues of the gastrointestinal tract [232]. The A33 glycoprotein has three structural domains: an extracellular region of 213 amino acids, a single hydrophobic transmembrane domain and a highly polar intracellular tail. The basic extracellular structure consists of domains similar to the IgG fold and the intracellular domain, including a quadruple cysteine repeat followed by a highly acidic sequence [234]. Its closest homologs comprise the Coxsackie adenovirus receptor (CAR), cortical thymocyte receptor (CTX), endothelial cell adhesion molecule (ESAM), junction adhesion molecules 1–3 (JAM) and CEA-related cell adhesion molecules (CEACAMs) [235,236,237]. The putative function of GPA33 is its role as an adhesion molecule. Moreover, GPA33 plays a role in trafficking proteins to vesicles. This intracellular function of GPA 33 depends on the particular phase of the cell cycle, with the peak in the G2/M phase. The lowest pGa 33 level was observed in the S phase, while mRNA levels were highest in the S phase but almost absent in the G1 phase [233]. 

PGA 33 is an immunomarker for CRC with a specificity of 85.4% and a sensitivity of 95.9%.

This biomarker had similar sensitivity to CDX2 but its specificity was higher [238]. 

### 3.4. Cadherin-17 (CDH17)

Cadherin-17 (CDH17) is a calcium-dependent transmembrane glycoprotein and a member of the cadherin superfamily [239,240,241]. This cadherin is transcriptionally regulated by CDX2 via binding to elements in the 5′ flanking region of the gene in normal, metaplastic and neoplastic tissues of the gastrointestinal tract [242,243]. CDH17 is expressed in the basolateral plasma membrane of enterocytes and goblet cells in normal tissue [240,244]. CDH17 occurs in cholesterol-rich fractions, where it is responsible for tissue integrity [245]. Cadherin-17 has been expressed in normal, small and large intestinal, pancreatic duct epithelium and in adenocarcinomas originating from gastric, pancreatic and colorectal tissue. Less than 1% of carcinomas outside the digestive system were positive for CDH17. Therefore, another name for this glycoprotein is liver–intestine cadherin [241]. CDH17 participates in cell–cell adhesion in the intestinal epithelium through interaction with α2β1 integrin [240]. CDH17 modulates integrin activation and signalling to induce specific focal adhesion kinase FAK/protein tyrosine kinase2, paxillin, RhoA, Rac and Ras, the activation of which leads to the induction of extracellular signal-regulated kinase and Jun N-terminal kinase, an increase in cyclin D1 and the proliferation of colon cancer cells in liver metastasis [245]. CDH17 silencing in KM12 cells suppressed tumor growth and liver metastasis after subcutaneous or intrasplenic inoculation in nude mice. Integrins are a family of cell adhesion molecules and are receptors for extracellular matrix proteins. α2β1 integrin interacts with two different ligands, collagen IV and CDH17, using two different binding sites. CHD17 has a tripeptide RGD site for integrin binding as well as for some other ligands, such as fibronectin or fibrinogen [246,247]. RGD regulates β1 integrin activation and an increase in focal adhesion kinase and ERK1/2 activation in colon cancer metastatic cells. In other words, in CDH17-positive CRC, integrin α2β1 binding to CDH17 is inhibited and thereby prevents integrin activation and in consequence metastasis, which may have therapeutic potential [248]. 

CDH17 is a useful immunohistochemical marker for the diagnosis of primary and metastatic colorectal adenocarcinomas with a specificity in the range of 50–83.8% and a sensitivity in the range of 96–100% [230,249,250]. CDH17 seems slightly more sensitive than CDX2 in detecting gastrointestinal adenocarcinomas. While comparing the usefulness of CDH17 in immunohistochemical diagnostics in gastrointestinal carcinomas with CDX2, it has been observed that almost all colon adenocarcinomas (99%) are CDH17-positive/CDX2-positive [243]. In turn, the combination of CDH17 and SATB2 served as potential optimal markers for the differential diagnostics of pulmonary enteric adenocarcinoma (PEAC) and metastatic colorectal adenocarcinoma with high sensitivity (76.92%) and specificity (100%) [251]. PEAC is a rare type of non-small cell lung cancer with similar histological and immunohistochemical morphology to colorectal adenocarcinoma. A positive correlation between high expression of CDH17 and liver metastasis and poor survival of CRC patients has been observed [245].

### 3.5. Cytokeratins 

The cytoskeletal framework consists of three kinds of cytoskeletal filaments called microfilaments, intermediate filaments and microtubules. Intermediate filaments (IFs) are responsible for organizing internal three-dimensional cellular structure and tension, giving shape to cells [252]. Moreover, IFs are the most chemically stable cellular structures, resisting high temperature, high salt and detergent solubilization [253]. All IFs have a dimeric central rod domain, which is a coiled structure consisting of two parallel α-helices flanked by head and tail domains. IFs are classified into five categories based on their rod domain amino-acid sequences. Type 1 IFs include the acidic keratins and are present in epithelial cells. Type 2 IFs include the basic keratins and are found in epithelial cells, too. Type 3 IFs include vimentin, desmin and glial fibrillary acidic protein. Type 4 IFs occur in neurofilaments. Type 5 IFs are the nuclear lamins [254]. 

Cytokeratins are intermediate filament-forming protein localized in the cytoplasmic cytoskeleton. They are characteristic of epithelial cells only. Cytokeratins regulate many cellular functions, such as cell size determination, apical–basal polarization, protein translation control, organelle positioning and membrane protein targeting [253,255,256,257,258,259]. Nowadays, keratins have been identified and divided into two groups. Type I includes 28 (20 epithelial and 11 hair) keratins and type II includes 26 (20 epithelial and 6 hair) [260]. Similar to other IFs, keratins contain a central coiled α-helical rod domain of about 310 amino acids, which is subdivided into subdomains (coils 1A, 1B, 2A and 2B). The subdomains are connected with three linkers, L1, L12 and L2. The non-helical head domains consist of subdomains V1 and H2. The tail domains, which are non-helical, also have subdomains, H2 and V2 [254]. Cytokeratins have been widely used as immunohistochemical markers in CRC diagnostics. 

#### 3.5.1. Cytokeratin 7 (CK7)

Cytokeratin 7 (CK7) is a type II member of the keratin superfamily. Despite widespread expression in epithelia, its role still remains unclear. CK7 expression was significantly associated with poor tumor differentiation and the extent of tumour budding. Disease progression was more frequently observed in CK7-positive cancer patients than in CK7-negative patients (52% versus 41%) [261]. 

CK7 is expressed in metastatic lymph nodes and is correlated with shorter overall survival and the presence of distant metastases at diagnosis. Interestingly, this correlation was not observed between CK7 expression in the primary tumour and overall survival. It seems that CK7 expression in metastatic lymph nodes in CRC patients is a poor prognostic factor [262].

#### 3.5.2. Cytokeratin 20 (CK20)

Loss of cytokeratin 20 (CK20) has been associated with older age (above 56 years), right-sided tumours, higher grade and mucinous histology, advanced stage, increased intertumoral lymphocytic infiltration (creating Crohn’s disease-like infiltrate) compared with CK20-positive tumours. Loss of CK20 is therefore correlated with lower disease-free survival and overall survival [263,264,265]. In turn, lack of staining or low expression of CK20 was significantly associated with poor differentiation, large tumour size and mismatch repair deficiency but was not found to significantly influence prognosis [266]. Low CK20 levels had been detected in association with high microsatellite instability [267]. 

#### 3.5.3. CK20+/CK7−

CK20 is specific for colon, urothelial and Merkel cell carcinoma. CK7 is characteristic of glandular malignancies originating in the breast, respiratory tract, biliary tract and Mullerian epithelium [253,268,269]. The CK20+/CK7− profile is expressed in about 75–95% of CRC cases [253,268,269,270,271]. Therefore, the CK20+/CK7− profile is characteristic of colon carcinoma and is a widely used diagnostic tool to determine the site of origin in metastatic carcinomas [266,272]. According to Al-Maghrabi et al., the most common profile was CK20+/CK7−, observed in 60.4% of CRC cases, and CK20−/CK7−, observed in 35.4% of cases. A mixed CK20+/CK7+ and CK20−/CK7+ profile was reported in 2.1%. Al-Maghrabi et al. did not note statistically significant correlations between CK20/CK7 immunohistochemical profiles and clinicopathological characteristics (such as age, sex, tumour size and location, lymph node status, etc.), prognosis and survival [272]. Other investigators revealed a positive correlation between a CK20+/CK7+ profile and an advanced stage of CRC [266,273]. In another study, the CK20−/CK7+ profile was characteristic of right-sided and high-grade colon cancer [274]. 

#### 3.5.4. Cytokeratin 15 (CK15)

Cytokeratin 15 (CK15) is a type I keratin lacking a defined type II partner and participates in maintaining cytoplasmic stability [275,276,277]. CK15 is present in the basal keratinocytes of stratified epithelia, while abnormal expression of CK15 is involved in tumorigenesis and cancer progression [278,279,280,281,282,283]. 

CK15 was found to be highly expressed in colorectal cancer tissue [284] and was correlated with a poorer prognosis. CRC patients with high CK15 expression had significantly lower overall survival compared with those patients with low CK15 expression. High CK15 expression positively correlated with the differentiation and staging of CRC. CK15 might be treated as an independent prognostic factor in CRC [284]. 

#### 3.5.5. Cytokeratin 18 (CK18)

Cytokeratin 18 (CK18) is upregulated in many types of human cancers (e.g., hepatocellular carcinoma, cervical carcinoma) and is correlated with clinical progression and worse prognoses [285,286]. CK18 participates in a diverse range of normal cellular processes, such as cell proliferation, the cell cycle, apoptosis, motility and cell signalling [287]. 

CK18 expression is increased in CRC cancer tissue in comparison with adjacent normal colorectal tissue. Moreover, high CK18 expression is positively correlated with advanced clinical stage, metastases to lymph nodes or other solid organs and poor differentiation. CK18 overexpression is an independent predictor of overall survival in CRC patients with upregulated CK18 expression in tumour tissue. Downregulation of CK18 expression inhibited CRC cell viability, migration and invasion in an in vitro study [288].

### 3.6. Telomerase 

Telomeres are regions protecting the ends of chromosomes and contain unique hexameric repeats (TTAGGG)_n._ They regulate chromosomal integrity and cell life span. When a critical shortened length is reached, cells begin to undergo replicative senescence [289,290]. Telomerase is a ribonucleoprotein enzyme complex whose activity helps cells to avoid senescence. Telomerase contains a telomere-specific reverse transcriptase (hTERT) which shows structural and functional similarities to viral transcriptases. The second component is an internal RNA (telomerase RNA-hTR) template sequence on which the telomeric repeats are synthesized [289,291]. 

Telomerase is present in immortalized cells, such as germ line cells, and 80–90% of human cancer cells [292]. 

The overexpression of hTERT increases replicative potential in CRCs and the risk of recurrence [289]. CRC patients with elevated expression of hTERT were found to have a significantly worse median overall survival compared with patients who had tumours characterized by low expression of hTERT (37 months vs. not reached), regardless of stage or systemic treatment given. The hazard ratio for death was 15 times higher in patients with metastatic disease and with elevated hTERT than in patients with low hTERT expression [289]. 

In other studies, it was observed that elevated hTERT expression was associated with a more advanced stage and a worse prognosis, patients with colorectal cancer having lower disease-free survival (DFS) and overall survival (OS) [293,294]. 

hTERT seems to be a biomarker for recurrence which can be used to monitor responses to systemic therapy. 

Telomere length is an independent prognostic factor in CRC. Cancers with mean telomere lengths less than 6.35 Kb have a better prognosis [295]. Noncoding telomeric repeat-containing RNAs (TERRAs) regulate the enzyme telomerase, influencing telomere length. Bae et al. revealed that 18p TERRA expression was significantly correlated with telomere length. In a multivariate analysis, it has been shown that 18p TERRA expression was a significant independent prognostic factor for disease-free survival in CRC patients [296]. Ayomamlitis et al. reported that a significantly higher level of telomerase activity was found in colon cancer tissue (especially in right-sided tumors vs. left-sided) than in rectal cancer tissue. The same increase of telomerase activity was observed in normal adjacent tissue in patients with colon cancer in comparison with those with rectal cancer. Colon cancers had higer hTERT activity than rectal cancers [297]. In another study it was observed that high telomerase activity was significantly correlated with a worse prognosis compared to cancers showing moderate or low telomerase activity [298]. Telomere length and hTERT expression in cancerous colon tissue were significantly correlated with histopathological features and overall survival. Cancer tissue had significantly shorter telomeres than surrounding normal mucosa. Longer telomeres were noted in advanced CRC (stage II–IV) compared to stage I tumors. Significantly lower hTERT expression levels were found in CRC tissue compared to adjacent normal mucosa [299,300]. 

However, telomerase activity does not always correlate with hTERT expression in colon cancer possibly because of the presence of hTERT in infiltrating lymphocytes in normal mucosa [301]. Therefore, the measurement of hTERT alone may overestimate the actual presence of telomerase within both normal and cancerous bowel epithelial cells [302]. 

Table 6 presents the results of published studies in which telomerase activity in colon cancer cases was evaluated.

The specificity of telomerase activity and hTERT was found to be excellent, but the sensitivity was relatively low [292] (see Table 6). 

Thanks to recent medical advances, telomere length measurement and analysis of telomerase expression have shown promise as useful molecular biomarkers for early CRC cancer detection and monitoring and for identifying patients with a poor prognosis [309]. 

## 4. Diagnostic Stool Biomarkers for Colorectal Cancer

### 4.1. Guaiac-Based Faecal Occult Blood Testing (gFOBT)

Guaiac-based faecal occult blood testing (gFOBT) has been the most used screening technique for colorectal cancer worldwide for many years. It is a cheap and non-invasive method [2]. This test is used to identify people with >10 mL rectal blood loss daily. Low sensitivity (50% for colorectal cancer and 20% for adenomas), especially in the early stages of colorectal cancer, and low acceptance have been the main disadvantages of this method [310]. The next significant limitation of gFOBT is its low specificity. False positive results occurred after taking nonsteroidal anti-inflammatory drugs and compounds with peroxidase properties (such as meat and fruit), while taking vitamin C in high doses gave false negative results. It is necessary to follow a special diet and to take stool samples three time in order to perform gFOBT, which is a big inconvenience for patients [311]. 

Burch et al. reported a meta-analysis of 59 studies concerning FOBT efficiency. Sensitivities for the detection of all neoplasms ranged from 6.2% to 83.3% [312]. 

There are different types of FOB test, such as chemical and immunochromatographic, which vary in terms of execution technique, sensitivity and specificity. 

Fluorescent long DNA (FL-DNA) is a non-invasive method which uses a qPCR assay in order to evaluate long fragments of stool DNA by quantitative amplification of specific targets of genomic DNA. In the case of DNA fragments longer than 200 bp, it correlates with high specificity of colorectal lesion detection. This test can be used with an immunochemical-based faecal occult blood test (iFOBT) of the same sample [313]. Randomized trials showed that the effectiveness of gFOBT as well as iFOBT holds for the detection of left-sided lesions only [314]. 

Interesting screening options are the combination of FOBT with other methods which increase the detection rate. 

Combining iFOBT and a faecal microRNA-106a (miRNA) test (FmiRT) was found to improve the sensitivity (70.9%) and specificity (96.3%) of CRC detection. One quarter of CRC patients with false-negative iFOBT results had positive FmiRNA tests [18]. On the other hand, another study suggested that the combination of FOBT and M2-PK tests increased sensitivity to 90% and specificity to 62% [315]. The tumour pyruvate kinase tumour (Tumour M2 PK) is a dimeric form of the glycosylation enzyme and belongs to type M2 pyruvate kinase. The enzyme is the catalyst of the last reaction stage of the glycolytic sequence and converts phosphoenolpyruvate into pyruvate, being responsible for the production of ATP within the metabolic pathway in which it participates. 

Ten-thousand asymptomatic participants aged 50–75 years were enrolled in the Prevention Project for Neoplasia of the Colon and Rectum (PRENEC) conducted in Chile. Participants with positive iFOBT results or a family history of CRC underwent a colonoscopy. The results of this screening procedure were compared with a previous national screening program (PREVICOLON). The adenoma detection rate was 26.7% in PREVICOLON and 41.8% in PRENEC. The cancer detection rate was 1.1% and 6%, respectively. A high return rate for samples and necessary contact with participants to invite them for colonoscopy are pivotal factors in CRC screening with iFOBT [316]. 

Nowadays, FOBT has been replaced by other screening tests, such as faecal immunochemical testing (FIT), multitarget stool (mt-sDNA), the FOBT DNA test and miRNA tests [317,318,319,320,321].

### 4.2. Faecal Immunochemical Test (FIT)

The faecal immunochemical test (FIT) is a modification of the guaiac-based FOBT, measuring the presence of blood degraded by digestive proteolytic enzymes [322]. The true positive rates of FIT were low for early stage CRC (in the range of 18–33%) and advanced adenomas (in the range of 9–19%) [323,324]. 

FIT has a wide range of sensitivities for all stages of CRC, starting at 25% and reaching as high as 79% [325,326]. Sensitivities of FIT were higher in cases of more advanced CRC: for T3, sensitivity was 83% (ranging from 68% to 91%), and for T1 was 40% (ranging from 21% to 64%) [327]. 

Interestingly, a meta-analysis of studies from Europe and Australia revealed that FIT CRC screening uptake was significantly lower (16%) in men aged 40–75 years than women. Additionally, women were found (non-significantly) to participate more often in FIT screening than men in North America and South America. A similar uptake of FIT-based screening was noted in Asian studies [328]. 

In a case–control study conducted on asymptomatic counting, over 23,000 participants of a Chilean population who had positive FIT results or a family history of CRC underwent a colonoscopy. According to this study, male sex, a positive family history, dietary habits (low intake of fibres and/or cereals) and older age (seventh and eighth decade of life) were the main risk factors of CRC [329]. 

### 4.3. Stool DNA (sDNA)

Stool specimens seems to be more suitable for early detection of CRC than blood samples due to the fact that exfoliating tumor cells appear in the large intestine or rectal lumen much earlier than the beginning of vascular invasion by tumor cells during colorectal carcinogenesis [132]. Tumor cells excreted with stools have DNA containing aberrant genetic and epigenetic alterations which are potential biomarkers for cancer diagnosis. 

The stool DNA test is the first noninvasive screening tool that targets both human hemoglobin and a panel of specific genetic alterations [330]. The Food and Drugs Administration approved multi-target stool DNA (mt-sDNA) for colorectal cancer screening in average-risk patients in 2014 [320,331,332,333]. There is a multi-target stool DNA test (Cologuard, a combination of *NDRG4* and *BMP3* DNA methylation, *KRAS* mutations, and haemoglobin) and a plasma *SEPT9* DNA methylation test (Epi proColon), which has been used more widely in clinical practice so far [120]. 

Many international oncological guidelines recommend mt-sDNA tests as an option for colorectal cancer screening. Stool-based DNA testing provides an entirely noninvasive population-based screening strategy which patients can perform easier than FOBT. 

The sensitivity of mt-sDNA testing for detection of CRC is as high as 90% in asymptomatic individuals [321,331,334,335]. The specificity with DNA testing ranges from 86.6% to 98% [334,336]. One-third of patients diagnosed in this way with CRC have advanced neoplasia [333]. Positive results in mt-sDNA tests correlate significantly with multiple lesions, larger precancerous lesions and lesions with tubulovillous architectures [337]. Moreover, in most cases, positive mt-sDNA is linked with a right-side location of CRC in colonoscopies [338]. According to Wang et al., there is no association between an sDNA test for SDC2 methylation and clinopathological features of CRCs, such as age, TNM stage, tumor location (colon vs. rectum) and tumor differentiation, except for gender [336]. 

Multi-target stool DNA tests have high positive predictive value. In case of a positive mt-sDNA test, a colonoscopy is the next step in the diagnosis of a colorectal neoplasm [338,339,340]. 

The sensitivity of detection of advanced adenomas and sessile serrated polyps measuring 1 cm or more in the greatest dimension with mt-sDNA is 42.4% [331,334,336]. 

A meta-analysis of 13 studies including over 700 patients using faecal gene methylation as a CRC screening method revealed that sensitivity for CRC detection was 78%, specificity was 90% and diagnostic objective response was 48%. The sensitivity for the detection of adenomas was 63% and the specificity was 93% [341]. 

Pickhardt et al. presented an interesting study in which an (FDA) approved mt-sDNA test was compared with CT colonography in asymptomatic individuals. Overall detection rates for advanced neoplastic lesions were significantly greater with CT colonography screening (5.0%) than with the mt-sDNA test (2.7%). Overall detection rates for CRC were 0.31% and 0.23%, respectively [342].

The main limitation of this method is false positive results (i.e., a positive mt-sDNA test and a negative high-quality colonoscopy performed after DNA testing) [334,338].

There are several meta-analyses assessing the diagnostic value of stool DNA testing [343,344,345,346,347,348]. Zhai et al. reported the pooled sensitivities for single- and multiple-gene stool DNA (methylation and mutation) tests in CRC to be 48.0% and 77.8%, respectively, and the pooled specificity for single- and multiple-gene assays to be 97.0% and 92.7%, respectively [344]. In another study (for methylated single- and multiple-gene tests in stool samples), Luo et al. demonstrated an overall sensitivity of 62% and 54%, respectively, and a specificity of 89% and 88% in CRC and adenoma patients, respectively [346]. Zhang et al. reported a sensitivity ranging from 51% for adenomas to 73% for CRCs and a specificity of 92% [345]. In another meta-analysis, the pooled sensitivity of the combined single- and multiple-gene DNA hypermethylation tests on stool specimens was 0.71 and specificity was 0.92 for [343]. In Mojtabanezhads et al.‘s meta-analysis, this method was shown to have a lower efficiency than reported in previous meta-analyses: the sensitivity for CRC and adenomas was 56.5% and 32.6%, respectively, and the specificity was 93.2% for CRC and adenomas [347]. In a published meta-analysis including results obtained for over 16,000 patients, it was revealed that the sensitivity of the stool methylation test with a single SDC2 gene was 83.1% and specificity was 91.2%, what is promising in case of this diagnostic method as employed instead of colonoscopy [348].

The most important studies of non-invasive, methylated DNA stool biomarkers used in colorectal cancer detection are collected in Table 7.

### 4.4. Faecal Immunochemical Test (FIT) and Stool DNA Test

Other diagnostic tools, such as DNA- or RNA-based tests, evaluated in a community-based population, were found to improve the efficiency of the FIT method [378].

The sensitivity of multi-target stool DNA (mt-sDNA) testing for the detection of CRC in asymptomatic individuals at average risk of cancer was significantly higher compared to FIT: 92% vs. 74%, respectively [331,334]. Similarly, the sensitivity for detecting advanced precancerous lesions was significantly higher for mt-sDNA testing (42.4%) than for FIT (23.8%) [334]. The specificity of the mt-sDNA test was 86.6% and the FIT had a specificity of 94.9% [335]. Carether et al.’s study has comfirmed that using mt-sDNA testing increased the detection of advanced adenomas and sessile serrated polyps (42% and 42%) compared to (24% and 5%) FIT. On the other hand, overall specificity for detection of all lesions was lower when using mt-sDNA (87%) than FIT (95%) [331]. Mu et al. have demonstrated that a DNA–FIT test increased the sensitivity of detection to 97.5% for CRC and to 53.1% for advanced adenomas [379]. The positive detection rate of mt-sDNA as well as FIT has been found to be independent of age, gender, tumor location and tumor size [380].

### 4.5. Methylation of DNA

Colorectal cancer is a heterogenous disease involving epigenetic alterations of the DNA, especially CpG island methylation, which occurs at a higher frequency than genetic mutations. Hypermethylation induces transcriptional silencing or downregulation of suppressor genes, what is one of the links to carcinogenesis. Epigenetic alterations are heritable changes in the structure or chemical composition of DNA or histone and other DNA-bound proteins affecting gene expression, but they do not result in changes to the DNA sequence [381]. The methylation of DNA involves the addition of a methyl group to the 5′ position of the pyrimidine ring of cytosines to produce 5-methylcytosine catalysed by DNA methyltransferases in CpG dinucleotide sequences localized in short CpG-rich DNA regions (CpG islands), centromeric repeats and rDNA [350,356,381]. Hypermethylation inactivates tumour suppressor genes at each step of carcinogenesis from polyps to colorectal adenocarcinomas. Many genes, such as *APC, MLH1*, *MGMT*, *SFRP1*, *SFRP2*, *CDK2A*, *TIMP3*, *VIM*, *SEPT*, *CDH1* and *HLTF*, are methylated in CRC, especially within the promoter region [382].

Quantitative DNA methylation analysis revealed that approximately 20% of colorectal cancers have a high frequency of promoter methylation (the CpG island methylator phenothype, CIMP). These CIMP cancers are associated with female sex, old age, proximal colon location, poor differentiation and harbouring of *MSI*, *KRAS* and *BRAF* mutations. There are two subclasses of CIMP tumours: CIMP-high tumours (or CIMP1) have *BRAF* mutations and are microsatellite-unstable; CIMP-low (or CIMP2) tumours have *KRAS* mutations [381].

Thousands of abnormally methylated CpG positions in CRC genomes are often located in non-coding parts of genes. A good example of such an alteration might be the *Syndecan-2* gene.

*Syndecan-2 (SDC2)* (called fibroglycan) belongs to the Syndecan family and encodes a transmembrane type I heparen sulfate proteoglycan. The SDC2 promotor region is frequently hipermethylated in colorectal cancer and has been detected in blood and stool samples from colorectal cancer patients [382,383,384]. The expression of SDC2 is upregulated by demethylation and inhibition of histone deacetylation in colorectal cancer cells with SDC2 methylation. In other words, the expression of SDC2 is suppressed by promoter methylation [384]. DNA methylation tests are not correlated with tumor differentiation, TNM stage, location, sex or age [336,385]. Faecal-originated DNA containing methylated *SDC2* might be a promising biomarker with high sensitivity (77.4–90.2%) and specificity (88.2–98%) for the noninvasive detection of colorectal cancer [336,384,386]. The very rare false positive results can be an effect of occasional elevated methylation status in healthy individuals [336]. On the other hand, methylated SDC2 in faecal samples was detected in 42–66.7% of advanced adenomas and 24.4% of non-advanced adenomas; therefore. this method might not be suitable for detecting precancerous lesions [336,384,385].

Colorectal cancer patients with hypermethylation of the secreted frizzled related protein 2 (SFRP2) gene in tumor tissue, stool and blood samples had a significantly lower overall survival than those with negative results for SFRP2 methylation analyses. Serum SFRP2 methylation was significantly correlated with poor differentiation grade, higher TNM stage with positive lymph node metastasis status and deeper tumor invasion in the bowel wall [387].

N-Myc downstream-regulated gene 4 (NDRG4) plays a role in cell growth and differentiation. NDRG4 is downregulated in colorectal cancer [357].

Septins are a group of scaffolding proteins existing in stable six to eight-subunit core heteromers. The octamer constitutes of two molecules of each of SEPT2, SEPT6, SEPT7 and SEPT9 subunits [388].

The SEPT9 gene encodes a GTP-binding protein involved in cell proliferation and migration, cytokinesis and angiogenesis [389]. Hypermethylation of the promoter region of the Septin 9 gene disturbs its transcription and is an important factor in carcinogenesis [115]. Aberrant methylation of this gene occurs in CRC tissues [132].

Bone morphogenetic protein 3 (BMP3) belongs to a transforming growth factor-beta (TGFβ) superfamily of cytokines. BMP3 binds to cell surface receptors and influences the regulation of the transcription of SMAD4 genes and, in consequence, inhibits growth. Downregulation of the BMP3 tumor suppressor gene might be involved in the early stages of CRC tumorigenesis [390].

### 4.6. Stool miRNA

miRNAs are small (18–25 nucleotide in length) noncoding RNAs (ncRNA) participating in post-transcriptional regulation of gene expression [332,391]. miRNAs are integrated into the RNA-induced silencing complex. This complex regulates the expression of target messenger RNAs (mRNAs) through post-transcriptional processing by binding primarily to the 3′ untranslated region (3′ UTR) of target mRNAs, which leads to the inhibition of translation or mRNA degradation [392,393]. ncRNAs are abnormally expressed in CRC and can behave as tumor suppressors (tsmiRs) or as oncogenes (oncomiRs), depending on the downstream target genes or pathways they regulate. It is interesting that the clinopathological features and prognostic factors of CRC correlate with the expression status of miRNAs [394,395].

The possibility of a distinction between the miRNA profile of CRC and normal bowel mucosa miRNAs has contributed to the development of a new screening tool, diagnostic and prognostic biomarkers. Obtaining samples from CRC tissue, plasma and stool is easy and reproducible [391]. There are some limitations to stool miRNA analysis. Firstly, faecal features, such as density, volume, etc., are changed daily, which makes the standardization of protocols difficult. Secondly, it is necessary to distinguish three sources of faecal miRNA: cell-free miRNAs from faecal homogenates, exosomal miRNAs from faecal exosomes and faecal colonocyte miRNAs [396].

The first report concerning the association between miRNA (miRNA-143 and miR-145) and CRC and precancerous lesions was published almost twenty years ago [394]. Currently, the list of tested miRNAs and their possible clinical applications is still being extended.

Noninvasive stool miRNA biomarkers for CRC detection are summarized in Table 8.

### 4.7. Faecal Bacteria

It is a well-known fact that large intestine microflora called microbiota have been implicated in the tumorigenesis of colorectal cancer [409,410]. The quantity of bacteria in the large intestine is 10^12^ cells per ml and cancer risk in this location is 12-fold greater than in the small intenstine, where the quantity of bacteria is 10^2^ cells per ml. These observations suggest that colon cancer might be a bacteria-related disease [409].

Depending on diet, drug intake and lifestyle, intestinal microbiota in healthy individuals vary in different countries [409]. Furthermore, the presence of blood in stools could influence the composition of the microbiome in the gut. Species such as *Bacteroides uniformis, Collinsella aerofaciens*, *Eggerthella lenta* and *Clostridium symbiosum* demonstrate increased abundance in patients with blood in their stools [411]. Tjalsma et al. proposed the hypothesis of carcinogenesis according to which some bacteria called ‘passengers’ (*Fusobacterium* spp., *Streptococcus gallolyticus* subsp. *gallolyticus, Clostridium septicum* and *Coriobacteriaceae* (*Slackia* and *Collinsella* spp.), the genus *Roseburia* and the genus *Faecalibacterium)* stimulate other bacteria called ‘drivers’ (such as *Bacteroides, Shigella, Citrobacter* and *Salmonella* or *E. coli)* which play a role in the first steps of colorectal cancer development [410]. Driver bacteria adhere to the epithelium of bowel mucosa and damage epithelial DNA, contributing to the initiation of CRC. It is interesting that the bacteria found in stools are not always the same bacteria as those attached to the epithelium. Moreover, bacterial drivers may disappear from neoplastic tissue as they are replaced by passenger bacteria [410]. Some individuals are carriers of a higher proportion of driver bacteria than others, which can be the result of genetic conditioning and lifestyle, and for which reason they might have a higher risk of CRC [409].

Therefore, alterations in microbiota are potentially good diagnostic biomarkers for CRC [412]. Special attention has been paid by researchers to *Fusobacterium nucleatum*, an anaerobic, gram-negative oral commensal, which is a pathogenetic factor of, e.g., inflammatory bowel disease and CRC [413,414,415].

An increase of *Fusobacterium nucleatum* in stools was associated with the presence of *Fusobacterium nucleatum* in the tumor tissue of colorectal cancer patients and had a positive correlation with tumor stage [416,417]. Moreover, the presence of *Fusobacterium nucleatum* was positively correlated with cholesteryl esters and sphingomyelin metabolite classes in stools [417]. There were no differences in stool microbiota between colorectal cancer and adenomas because *Fusobacterium nucleatum* infiltrate different cells in the same way and activate their growth using FadA adhesin, which binds to E-cadherin, influencing oncogenic responses [417,418].

In another study, Suehiro et al., using droplet digital polymerase chain reaction (PCR), showed a higher number of copies of the *Fusobacterium nucleatum* genome in CRC advanced adenomas and carcinomas in situ in CRC patients’ stools compared to healthy individuals. This could indicate that *Fusobacterium nucleatum* is involved in the early stages of colorectal tumorigenesis [419]. In the next study, *Fusobacterium nucleatum* were detected in CRC patients’ stools in higher levels than in patients with dysplasia and controls with a sensitivity of 69.2% for CRC [420]. Similarly, DNA of colibactin-producing bacteria (clbA) was found to be present significantly more often in samples isolated from the stools of patients with CRC compared to patients with dysplasia or healthy volunteers. The sensitivity of an individual marker for clbA1 + bacteria was 56.4%. The specificity of both assays was close to 80% [420].

The ratio of *Fusobacterium nucleatum* to *Bifidobacterium* probiotic (Fn/Bb) proved to have a high sensitivity (84.6%) and specificity (92.3%) in detecting CRC. The combination ratio of Fn/Bb and *Fusobacterium nucleatum* to *Faecalibacterium prausnitzii* (Fn/Fp) increased sensitivity to 90% in detecting CRC in stage I [421].

The detection of alterations in intestinal abundance of four faecal bacteria, *Fusobacterium nucleatum, Bacteroides clarus*, *Roseburia intestinalis* and *Clostridium hathewayi*, in combination with FIT improved the sensitivity for CRC detection to 92.8% and the specificity to 81.5% [422]. Bacterial markers with FIT were correlated with stage I, II and III CRC but not stage IV [422].

In Wong’s et al. study, a 132-fold higher abundance of *Fusobacterium nucleatum* in CRC patients and a 3.8-fold increase was observed in patients with advanced adenomas. The combination of FIT and *Fusobacterium nucleatum* markers resulted in improved detection rates for CRC with a sensitivity of 92.3% and a specificity of 93.0%, without significant difference between cancer stages. In turn, the sensitivity was 38.6% and the specificity was 89.0% for advanced adenomas [423].

Clos-Garcia et al. observed a high level of *Parvimonas* anaerobic bacteria in CRC stools’ microbiota, which was associated with the increased activation of methane-related pathways [417]. In turn, *Adlercreutzia* bacteria, producing equol from isoflavonoids consumed in the diet, were more numerous in adenoma patients’ feces and could be used as an early CRC biomarker [417].

Using *Clostridum symbiosum* in the diagnosis of CRC is a very interesting option. It is still unclear if *Clostridium symbiosum* is a cause or a result of CRC. Xie et al. showed that the abundance of *Clostridium symbiosum* in stools was positively correlated with both early and advanced stages of CRC and advanced adenomas. The sensitivity of this method was approximately two-fold higher compared to FIT tests both for CRC and adenomas. The combination of *Clostridium symbiosum* detection and FIT increased sensitivity by 4–24%, and nearly one-third more patients were diagnosed with early-stage CRC [424].

### 4.8. The Gut Microbiota and miRNA

The molecular interactions between host and gut microbiota are mediated by proteins, metabolites and small RNA (sRNA), human micro RNA (hsa miRNA) and human small noncoding RNA (hsa sncRNA) [425]. It is a well-known fact that *Fusobacterium nucleatum* and Epstein–Barr viral sRNA are upregulated in CRC tissue but not in adjacent normal mucosa [426]. Futhermore, *Fusobacterium nucleatum* and *Escherichia coli* can pick up and incorporate faecal miRNAs, which modulate bacterial gene transcripts and influence bacterial growth [427]. An encouraging result of Tarallo’s et al. study was the extraction of a human and microbial sRNA signature which can be used to improve diagnosis of CRC by combined analyses of metagenomic and small RNA sequencing (sRNA-Seq) data.

hsa-miR-21-5p, hsa-miR-200b-3p, hsa-miR-1290-5p, hsa-miR-4792-3p and hsa-miR-1246-3p were found to be significantly upregulated in CRC cases; therefore, the five hsa-miRNAs could represent a new class of biomarkers for this disease. The differential expression analysis showed that hsa-miR-30-5p might be a good candidate biomarker for adenomas [425].

## 5. Volatile Organic Compounds (VOC)

Cancer metabolomes are defined as cancer-specific metabolites of low molecular weight below 1500 Da. Some of these metabolites are VOCs—organic chemicals that have a high vapour pressure at room temperature. Their production and release might be altered in some diseases, such as cancer, and could therefore be used in their detection. Genetic and proteinic changes contributing to the peroxidation of the cell membrane might thereby create CRC-specific VOCs [428]. In addition, the alterations in gut microbiota influence directly the profile of VOC production [429,430,431].

### 5.1. Urinary VOCs

Urinary biomarkers could be an option for non-invasive colorectal cancer screening instead of stool testing due to the fact that urine is simple to collect and easy to store. Additionally, this procedure is reproducible and has high patient acceptability.

Several studies utilised a mass spectrometry technology named field asymmetric ion mobility spectrometry (FAIMS) and gas chromatography coupled with ion mobility spectrometry (GC–IMS) as techniques for detecting VOCs in urine [432,433].

The sensitivity ranged from 69% to 88% with 60% specificity for urinary VOCs, using FAIMS technology [432,434]. In a very interesting study, Widlak et al. observed that the additional detection of urinary VOCs in the FIT false-negative (i.e., failed to detect cancer) increased the detection of CRC. Using both faecal (FIT and faecal calprotectin) and urinary VOC detection increased diagnostic accuracy for CRC from 80% to 97% and specificity to 72% in patients with lower gastrointestinal symptoms [435].

Mozdiak et al. tested the urinary VOCs from FOBT-positive patients using FAIMS and GC–IMS. The ability to distinguish CRC from healthy controls using FAIMS and GGC–IMS was high, with an AUC of 0.98 and 0.82, sensitivity of 100% and 80% and specificity of 92% and 83%, respectively. Urinary VOCs demonstrate a high sensitivity for adenoma detection. Disadvantages, such as the lack of specificity and high false positive rate, do not allow this method to be used as a standard for adenoma detection [433].

### 5.2. Stool VOCs

Based on current data analysis, faecal VOCs seems to have high diagnostic value for CRC and adenoma screening. All the papers have reported increased levels of amino acids and short chain fatty acids and decreased levels of bile acids and polyol alcohols in the gas phase of faeces [436].

Faecal VOCs from individuals who had a positive FOBT were analysed by selected ion flow tube mass spectrometry (SIFT-MS). Ions most likely originated from hydrogen sulphide, dimethyl sulphide and dimethyl disulphide reach statistically significantly higher levels in samples from high-risk compared to low-risk subjects. Results using multivariate methods show that the test gives a correct classification of 75%, with 78% specificity and 72% sensitivity on FOBT-positive samples, offering a potentially effective alternative to FOB [431].

Hydrogen sulphide is one of the stool VOCs responsible for the development of CRC.

Hydrogen sulphide is produced by endogenous enzymatic reactions in the bowel and microbiota in the gut. Lower levels of hydrogen sulphide are normal but higher levels (above 2.4 mmol/kg) are toxic. Its non-competitive binding to cytochrome c oxydase inhibits the binding of oxygen to cytochrome c oxydase, leading to the reduction of cellular adenosine triphosphate (ATP).

Hydrogen sulphide plays a role in the function of the ATP-sensitive potassium channels, the activation of which in turn influences the balance of the biological effects of hydrogen sulphide [431]. Higher levels of hydrogen sulphide in the lumen and faeces disturb the balance of microbiota. This phenomenon occurs in, e.g., CRC patients. Additionally, hydrogen sulphide leads to DNA damage which causes CRC [431].

Bond et al. identified a three-faecal VOC panel consisting of propan-2-ol derived from acetone, hexan-2-one and ethyl 3-methyl- butanoate, arising as the product of reaction between ethanol and 3-methylbutaninoic, the presence of which is positively correlated with CRC diagnosis [437].

In another study, faecal gas compounds from CRC patients were analysed using gas chromatography. The amount of hydrogen was significantly lower in patients with T3–T4 tumors and advanced stage disease than in other CRC patients, showing 90% sensitivity, 57.7% specificity and 75% accuracy of detection. The amount of methyl mercaptan, sulphur-containing gas, was significantly higher in CRC patients. Increased amounts of methyl merceptan could be produced by *Fusobacterium nucleatum* from L-methionine by L-methionine-a-deamino-c-mercaptomethane-lyase [438]. A second hypothesis assumes that methyl mercaptan is an effect of a reaction between sulphur-containing amino acids with glucose or lactic acid. In turn, lactic acid is produced by cancer cells (the Warburg effect) [439].

### 5.3. Breath VOCs

Exhaled breath changes its chemical composition depending on the health or state of disease. Sampling of breath is non-invasive and simple; however, only a few breath tests are sufficient. Cancer-specific VOCs do not appear normally in exhaled breath and can be used for detecting CRC.

Altmore at al. showed that the initial specificity of gas chromatography–mass spectrometry (GC–MS) through the identification and quantification of VOCs in exhaled breath was 83%, with a sensitivity of 86% and an overall accuracy of 85% [439].

In the next study, the breath samples were assessed with GC–MS, which identified four VOCs: acetone, ethyl acetate, ethanol and 4-methyloctane. Levels of acetone and ethyl acetate were higher in CRC; in turn, levels of ethanol and 4-methyloctane were lower in the CRC group. The sensor technology (based on a breath pattern to identify different groups) allowed CRC patients to be distinguished from healthy individuals with 85% sensitivity, 94% specificity and 91% accuracy. Sensitivity was 100% in distinguishing advanced adenomas and healthy individuals using the sensor technique; specificity and accuracy were 88% and 94%, respectively [440].

Wang et al. used phase microextraction/gas chromatography–mass spectrometry (SPME/GC–MS) to analyse exhaled VOCs. In CRC patients, VOC samples contained significantly higher levels of cyclohexanone, 2,2-dimethyldecane, dodecane, 4-ethyl-1-octyn-3-ol, ethylaniline, cyclooctylmethanol, trans-2-dodecen-1-ol and 3-hydroxy-2,4,4-trimethylpentyl 2-methylpropanoate, but significantly lower levels of 6-t-butyl-2,2,9,9-tetramethyl-3,5-decadien-7-yne [441].

According to current data analysis for VOCs in the gas phase of excreted materials, breath VOCs seem to present an innovative and promising approach to CRC screening. However, a lack of a unified technique, a long list of substrates used for diagnosis and the low availability of this method are the main disadvantages of this potential diagnostic tool [434].

## 6. European Society of Medical Oncology (ESMO) Recommendations for Screening

The ESMO recommends that adults aged 50 or older until the approximate age of 74 years with an average risk of CRC should carry out a complete colonoscopy with an optimal repetition interval for a negative test every 10 years. Flexible sigmoidoscopy undergone every 5–10 years may be an alternative for those who refuse colonoscopy. In order to reduce the risk of a right colon neoplasm, it is recommended that a combination of an endoscopy with a faecal occult blood test be carried out annually [442].

Other invasive (i.e., capsule colonoscopy) tests are not recommended for screening.

A non-invasive test (FIT) is recommended in average-risk adults from the age of 50 not already taking part in colonoscopic screening programmes, with the optimal frequency being every year and no later than every three years. When the test result is positive, a colonoscopy is recommended [442].

Other non-invasive tests, such as mtDNA-based tests or tests using other markers (e.g., M2-PK), should also be performed [442].

### Colrectal Cancer Screening and the COVID-19 Pandemic

The COVID-19 pandemic has contributed to the delay and/or cancelation of many important diagnostic procedures, such as colonoscopies for colorectal cancer screening.

Therefore, alternative diagnostic methods meeting requirements that can be introduced instead of demanding complex medical procedures and intensive involvement of medical personnel should be sought.

Faecal DNA tests have a potential to be useful screening tools in place of colonoscopies [443,444]. In Gachabayovs et al.’s published meta-analysis, encouraged by their results, the authors suggest that stool methylation tests with the single SDC2 gene is a promising diagnostic method instead of colonoscopy in the COVID-19 pandemic [351].

Analysis of results from over 16,000 patients revealed that the sensitivity of the stool methylation test with the single SDC2 gene was 83.1% and that the specificity was 91.2%, showing it to be a promising diagnostic method instead of colonoscopy [348].

The next screening solution that could be conducted safely during the COVID-19 pandemic is colon capsule endoscopy (CCE), an innovative technique for evaluating mucosa of the colon and an interesting screening option [445]. Among individuals with one or more adenomas 6 mm or larger, specificity was 82% and sensitivity in those cases was 88% [446]. On the other hand, the high cost of the procedure, a lack of experienced physicians, the risk of capsule retention, obstruction and possible bowel perforation are significant limitations to CCE [447].

## 7. CRC Screening Today and Future Challenges

Better understanding of the molecular genetics and epigenetics of colon polyps and CRC has led to the development of molecular marker assays for CRC screening. Approaches to CRC detection reviewed in this paper are undoubtedly impressive in terms of numbers but often disappointing in outcomes due to the fact that “ideal” screening or diagnostic biomarkers should be highly sensitive and specific, easy to perform, cheap and commonly available. Therefore, the combination of the FIT and colonoscopy has still remained the strategy of choice in CRC screening across the world. Screening methods based on blood testing were supplemented by a new biomarker, methylated gene *septin9* (mSEPT9), in the last few years after approval by the FDA. It is worth recalling doubts related to use of this test in detecting precancerous lesions, however [125]. Currently, screening guidelines are being updated.

A challenge for the future is the standardization of protocols, such as extraction and quantification methods, as well as normalization techniques. Secondly, it is necessary to translate biomarkers to the clinical context. Thirdly, the elimination of sophisticated laboratory equipment would be optimal, with simple implementations that are not expensive and not time-consuming.

Let us hope that existing progress in CRC biomarker research will result in the development of new non-invasive tests in CRC screening and thereby the prevention of this disease.

## Figures and Tables

**Table 1 ijms-23-00852-t001:** According to Vacante in modification: screening/diagnostics liquid biopsy biomarkers in CRC [10].

Author/Year	Detection Method/ Biomarkers	Sensitivity [%]	Specificity [%]	HR/OS/ *p* Value
Tsai/2019 [13]	CellMax biomimetic platform (CMx)/CTC	Precancerous lesions: 76.6 CRC: Sn 86.9	Precancerous lesions: 97.3 CRC: Sp 97.3	
Flamini/2006 [14]	qPCR/ctDNA	ctDNA alone: 81.3 ctDNA + CEA: 84.0	ctDNA alone: 73.3 ctDNA + CEA: Sp 88.0	
Sun/2019 [15]	Epigenomics AG for Epi proColon 2.0/mSEPT9 DNA	Precancarous lesions: 17.1 CRC: Sn 73.0	Precancerous lesions: 94.5 CRC: 94.5	
Link/2010 [16]	TaqMan qRT-PCR */ fecal miRNAs		Increased expression of miR-21 and miR-106a in CRC and adenomas vs. normal controls (*p* < 0.05)	*p* < 0.05
Wang/2017 [17]	real-time PCR/ Serum miR-31, miR-141, miR-224-3p, miR-576-5p, and miR-4669		AUC = 0.995 (microarrays) AUC = 0.964 (double-blind validation test)	
Koga/2013 [18]	real-time RT-PCR/ fecal miR-106a	FmiRT: 34.2 iFOBT + FmiRT: 70.9	FmiRT: 97.2. iFOBT + FmiRT: 96.3	
Sazanov/2017 [19]	real-time qRT-PCR */plasma and saliva miR-21	plasma: 65 saliva: 97	plasma: 85 saliva: 91	
Yan/2018 [20]	qRT-PCR*/exosomal miR-6803-5p			OS: HR 2.93 (95% CI 1.35–6.37, *p* < 0.007) DFS: HR 3.26 (95% CI 1.56–6.81, *p* < 0.002) AUC = 0.7399
Peng/2018 [21]	real-time qPCR */ exosomal miR-548c-5p			OS: HR 3.40 (95% CI 1.02–11.27, *p* = 0.046)
Liu/2016 [22]	qRT-PCR */ exosomal lncRNA CRNDE-h	70.3	94.4	
Liu/2018 [23]	qRT-PCR */ exosomal miR-27a and miR-130a	miR-27a: AUC = 0.773 75 in the training phase, AUC = 0.82 80.0 in the validation phase, AUC = 0.746 80.0 in the external validation phase miR-130a: AUC = 0.742 82.5 in the training phase, AUC = 0.787 70.0 in the validation phase, AUC = 0.697 70.0 in the external validation phase miR-27a + miR-130a: training phase AUC = 0.846 82.5, validation phase AUC = 0.898, 80.0 and external validation phase AUC = 0.801 80.0	miR-27a: AUC = 0.773 77.5 in the training phase, AUC = 0.82 77.5 in the validation phase, and AUC = 0.746 77.5 in the external validation phase miR-130a: AUC = 0.742 62.5 in the training phase, AUC = 0.787 80.0 in the validation phase, AUC = 0.697 80.0 in the external validation phase miR-27a + miR-130a: training phase AUC = 0.846 75, validation phase AUC = 0.898, 90.0 and external validation phase AUC = 0.801 90.0	

* qRT-PCR quantitative reverse-transcription polymerase chain reaction.

**Table 2 ijms-23-00852-t002:** According to Liu and Li dysregulated miRNAs in colorectal adenoma and carcinoma [66,68].

Author/Year	Change	miRNA
Yin/2016 [69]	upregulation	miR-18a, miR-18b, miR-31, miR-142-5p, miR-212
Uratani/2016 [70]	upregulation	miRNA-21, miRNA-92a, miRNA-135b
Imaoka/2016 [71]	upregulation	miRNA-1290
Ho/2015 [72]	upregulation	miRNA-486
De Groen/2015 [73]	upregulation	miRNA-15a, miRNA-17. miRNA-20a
Wu CW/2014 [74]	upregulation	miRNA-31, miRNA-135 b
Wu CW/2012 [75]	upregulation	miRNA-92a
Tsikitis/201 [76]	downregulation	miRNA-143, miRNA-145, miRNA -30a
Tadano/2016 [77]	downregulation	miRNA-320 family
Yin/2016 [69]	downregulation	miR-145, miR-451, miR-638
Chen T/2017 [78]	downregulation	miRNA-137
Ho/2015 [72]	downregulation	miRNA-30
Hibino/2015 [79]	downregulation	miRNA148a
Fang/2015 [80]	downregulation	miRNA24, miRNA-320a, miRNA-423-5p

**Table 3 ijms-23-00852-t003:** According to Loktionov in modification miRNA biomarkers used for CRC detection [1].

Author/Year	Sample Type	Biomarker/Detection Method	Sensitivity [%]	Specificity [%]
Kanaan/2013 [88]	plasma	miR-532-3p, miR-331, miR-195, miR-17, miR-142-3p, miR-15b, miR-532, miR-652	polyps from controls [area under curve (AUC) = 0.868 (95% confidence interval [CI]: 0.76–0.98)]. stage IV CRC from controls with an [AUC = 0.896 (95% CI: 0.78–1.0)]. Receiver-operating-characteristic curves of miRNA panels for all CRC versus controls and polyps versus all CRC AUC values of 0.829 (95% CI: 0.73–0.93) and 0.856 (95% CI: 0.75–0.97)	
Giraldez/2013 [89]	plasma	miRNA-18a, miRNA-19a, miRNA-19b, miRNA-15b, miRNA-29a, miRNA-335/ up-regulated	areas under the receiver operating characteristic curve (AUROC) ranging from 0.80 (95% confidence interval [CI], 0.71–0.89) to 0.70 (95% CI, 0.59–0.80)	
Wang/2014 [90]	serum	miRNA-21, let-7g, miRNA-31, miRNA-92a, miRNA-181b, miRNA-203/ up-regulated	93	91
Slaby/2016 [91]	plasma	miR-20a/upregulated miR-155/upregulated	46 58.2	73.4 95
Slaby/2016 [91] Carter/2017 [92] Chen/2019 [93] Sabry/2019 [94]	plasma/serum	miR-21/upregulated	65–91.4	74.4–95
Carter/2017 [92]	plasma	miR-24/downregulated miR-29a/upregulated miR-92/upregulated miR-29b/downregulated miR-106a/upregulated miR-194/downregulated miR-200c/upregulated miR-320a/downregulated miR-372/upregulated miR-375/downregulated miR-423-5p/downregulated miR-601/downregulated miR-760/downregulated	78.4 69% 89% 61.4–77 74 72 64.1 92.8 81.9 76.9 91.9 69.2 80	83.8 89.1 70 72.5–75 44.4 80 73.3 73.1 73.3 64.6 70.8 72.4 72.4
Slaby/2016 [91] Carter/2017 [92]	plasma/serum	miR-92a/upregulated miR-96/upregulated miR-221/upregulated	65.5–74 65.4 86	71.2–82.5 73.3 41
Ng/2017 [95]	serum	miR-139-3p/downregulated	96.6	97.8
Wang/2017 [96]	serum	miR-139a-5p/upregulated	76.7	88
Liu/2018 [97]	plasma	miR-182/upregulated	78	91
Bilegsaikham/2018 [98]	serum	miR-196b/upregulated miR-338-5p/upregulated	63 76.30	87.4 92.50
Carter/2017 [92] Sabry/2019 [94]	serum	miR-210/upregulated	74.6–88.6	73.5–90.10
Krawczyk/2017 [99]	plasma	miR-506/upregulated miR-4316/upregulated	76.8 76.8	60.7 75
Imaoka/2016 [71]	serum	miR-1290/upregulated	70.1	91.2
Nonaka/2015 [100]	serum	miR-103/upregulated miR-720/upregulated	55.9 58.3	75 56.3
Sarlinova/2013 [101]	whole blood	miR-21/upregulated miR-221/upregulated miR-150/downregulated	80 (three markers)	74 (three markers)
Chang/2016 [102]	plasma	miR-92a/upregulated miR-223/upregulated	0.75 0.707 (AUC values)	
Slaby/2016 [91] Carter/2017 [92]	serum	miR-21 and miR-92a/both upregulated	68 (whole panel)	91.2 (whole panel)
Slaby/2016 [91] Carter/2017 [92]	plasma	miR-29a and miR-92a/both upregulated	83 (whole panel)	84.7 (whole panel)
Nikolaou/2018 [103] Carter/2017 [92]	plasma	miR-200c and miR- 18a/both upregulated	84.6 (whole panel)	75.6 (whole panel)
Slaby/2016 [91]	plasma	miR-223 and miR- 92a/both upregulated	76 (whole panel)	71 (whole panel)
Liu/2019 [104]	plasma	miR-320d/downregulated miR-1290/upregulated	81.2 (whole panel)	90.7 (whole panel)
Carter/2017 [92]	plasma	miR-431 and miR- 139-p3/ both upregulated	91 (whole panel)	57 (whole panel)
Slaby/2016 [91] Carter/2017 [92]	plasma	miR-601 and miR-760/both downregulated	83.3 (whole panel)	69.1 (whole panel)
Carter/2017 [92]	plasma	miR-19a, miR-19b and miR-15b/all upregulated	78.6 (whole panel)	79.2 (whole panel)
Nikolaou/2018 [103] Carter/2017 [92]	plasma	miR-24, miR-320a and miR-423-5p/ all downregulated	92.8 (whole panel)	70.8 (whole panel)
Slaby/2016 [91] Carter/2017 [92]	serum	miR-145/downregulated, miR-106a and miR-17-3p/upregulated	78.5 (whole panel)	82.8 (whole panel)
Slaby/2016 [91] Carter/2017 [92]	plasma	miR-409-3p/upregulated miR-7 and miR-93/ downregulated	82 (whole panel)	89 (whole panel)
Wikberg/2018 [105]	plasma	miR-18a, miR-21, miR-22 and miR-25/ all upregulated	67 (whole panel)	90 (whole panel)
Nikolaou/2018 [103]	serum	miR-23a-3p, miR-27a-3p, miR-142-5p and miR-376c-3p/all upregulated	89 (whole panel)	81 (whole panel)
Carter/2017 [92]	plasma	miR-29a, miR-92a/ upregulated, miR-601, miR-760/ downregulated	83.3 (whole panel)	93.1 (whole panel)
Chen/2019 [93]	serum	miR-21, miR-29, miR-92, miR-125, miR-223/all upregulated	84.7 (whole panel)	98.7 (whole panel)
Herreros-Villanueva/2019 [106]	plasma	miR-19a, miR-19b, miR-15b, miR-29a, miR-335, miR-18a/ all upregulated	91 (whole panel)	90 (whole panel)
Slaby/2016 [91]	plasma	miR-21, let-7g/ upregulated, miR-31, mir-92a, miR-181b, miR-203/ downregulated	96 (whole panel)	81 (whole panel)
Zhang/2019 [107]	plasma	miR-103a-3p, miR- 127-3p, miR-151a-5p, miR-17-5p, miR- 181a-3p, miR-18a-5p, miR-18b-5p/all upregulated	76.9 (whole panel)	86.7% (whole panel)
Liu/2018 [23]	plasma	exosomal miR-27a, miR-130a/both upregulated	82.5 (whole panel)	75 (whole panel)
Tian/2019 [108]	plasma	hsa_circ_0004585/upregulated	85.1%	51.1%
Marcuello/2019 [25]	plasma	miR-15b-5p/upregulated miR-18a-5p/upregulated miR-29a-3p/upregulated miR-335-5p/upregulated miR-19a-3p/upregulatedmiR-19b-3p/upregulated	81 (whole panel with fecal hemoglobin)	78 (whole panel with fecal hemoglobin)
Karimi/2019 [109]	plasma	miR-23a/upregulated miR-301a/upregulated	0.89 0.84 (AUC values)	
Tan/2019 [110]	plasma	miR-144-3p, miR-425-5p, and miR-1260b	93.8 (whole panel)	91.3 (whole panel)
Maminezdah/2020 [111]	serum	miR-143/downregulated miR-145/downregulated miR-19a/upregulated miR-20a/upregulated miR-150/upregulated let-7a/upregulated	0.76 0.78 0.87 0.83 0.75 0.71 (area under the ROC curves)	
Liu/2020 [112]	plasma	exosomal miR-139-3p/ downregulated miR-139-3p and CEA	0.726 (AUC value) 0.868 (AUC value)	
Jin/2020 [113]	serum	miR-4516/upregulated miR-21-5p/downregulated both of them	94.4 90.63 92.11	89.8 86.2 87.9

**Table 4 ijms-23-00852-t004:** Sensitivity and specificity of the *SEPT9* gene methylation assay for colorectal cancer detection.

Author/Year	CRC/Number of Cases	Assay Used	Sensitivity [%]	Specificity [%]
Grützmann/2008 [124]	252/354	research assay	72	90
Lofton-Day/2008 [125]	150/350	research assay	52	95
DeVos/2009 [126]	97/172	mSEPT9 assay	72	93
He/2010 [127]	182	research assay	75	96.47
Tanzer/2010 [128]	73/128	research assay	73	91
Herbst/2011 [129]	45/345	research assay	46.6	81.3
Warren/2011 [130]	50/144	Epi proColon 1.0	90	88.3
Toth/2012 [131]	92/184	Epi proColon 2.0	95.6 79.3	84.8 98.9
Alquist/2012 [132]	30/52	Epi proColon 1.0	39	79
Lee/2013 [133]	101/197	mS9 Colorectal Cancer Assay System	36.6	90.6
Church/2014 [122]	53/1516	Epi proColon 1.0	48.2	91.5
Potter/2014 [134]	44/1544	Epi proColon 1.0	68	80
Su/2014 [135]	172/234	MSP-DHPLC	88.4	93.5
Johnson/2014 [136]	101/200	Epi proColon 1.0	73.3	81.5
Jin/2015 [137]	135/341	Epi proColon 2.0	74.8	87.4
Kang/2014 [138]	80/132	Epi proColon 2.0	79.5	98.1
Toth/2014 [139]	34/84	Epi proColon 2.0	82.8	91.7
Song 2016 [121]	369/1133	Epi proColon 2.0	58–82.4 *	82–98.8 *
Ørntoft/2015 [140]	150/150	Epi proColon 1.0	73	82
Behrouz Sharif/2016 [141]	45/45	MS-HRM assay	84.4	99
Wu/2016 [142]	291/1031	Epi proColon 2.0 new SEPT9 assay	73.0 76.6	97.5 95.9
Nian/2016 [143]	2975/6952	Epi proColon 2.0	71	92
Fu/2018 [144]	98/558	Epi proColon 2.0	61.22	93.7
Xie/2018 [145]	123/248	research assay	61.8	89.6
Arellano/2020 [146]	10/10	Epi proColon 2.0	88.9	100
Hariharan/2020 [118]	7629	mSEPT9 test	69	92
Liu/2020 [147]	38/124	Epi proColon 2.0	85.6	90.1

* using different algorithm.

**Table 5 ijms-23-00852-t005:** According to Loktionov in modification: long non-coding RNA biomarkers used for CRC detection [1].

Author/Year	Sample Type	Biomarker(S)	Sensitivity [%]	Specificity [%]
Zhao/2015 [174]	serum	CCAT1 and HOTAIR	84.3	80.2
Wang/2016 [175]	serum	LOC285194, RP11-462C24.1 and Nbla12061	68.3	86.9
Dong/2016 [148]	serum	BCAR4, two mRNAs: KRTAP5-4 and MAGEA3	93.6	85.7
Dai/2017 [176]	serum	BLACAT1	83.3	76.7
Barbagallo/2018 [177]	serum	UCA1	100	43
Liu/2019 [178]	plasma	91H, PVT-1 and MEG3	82.8	78.6
Abedini/2019 [179]	plasma	ATB, CCAT1	82.0	75.0
Nikolaou/2019 [103]	whole blood	NEAT1 variant 1	69.0	79.0
Nikolaou/2019 [103]	whole blood	NEAT1 variant 2	70.0	96.0

**Table 6 ijms-23-00852-t006:** According to Chen and Chen in modification: telomerase activity in colon cancer [292].

Author/Year	Material	Colon Cancer [%]
Engelhardt/1997 [303]	colon tissue	90
Yoshida/1997 [304]	colon tissue	92
Myung/2005 [305]	colon tissue	97
Tatsumato/2000 [298]	colon tissue	96
Kawanishi-Kabata/2002 [306]	colon tissue	80
Myung/2005 * [305]	colon tissue	94
Fang/1999 [307]	colon biopsy	88.5
Yoshida/1997 [304]	colon washing	60
Ishibashi/1999 [308]	colon washing	58
Myung/2005 [305]	colon washing	62

* detection of hTERT.

**Table 7 ijms-23-00852-t007:** According to Loktionov in modification: non-invasive, methylated DNA stool biomarkers used in colorectal cancer detection [1].

Author/Year of Publication	Marker Type/Method	Stool Biomarker	Sensitivity [%]	Specificity [%]
Muller/2004 [349]	DNA methylation	SFRP2 methylation	training set: 90 independent test set: 77	training set:77 indepedent test set: 77
Petko/2005 [350]	DNA methylation	CDKN2A and MGMT methylation	CDKN2A:50 MGMT: 71	
Huang/2007 [351]	DNA methylation	SFRP2 methylation	CRC: 94.2 52.4 advanced adenomas: 52.4	93
Itzkowitz/2007 [352]	DNA integrity assay (DIA)	Vimentin methylation	vimentin methylation: 72.5 vimentin methylation + DIA: 87.5	vimentin methylation: 86.9 vimentin methylation + DIA: 82
Wang/2008 [353]	DNA methylation	SFRP2 methylation	CRC: 87, advanced adenomas: 61, hyperplastic polyps: 42.3 overall: 76.8	
Itzkowitz/2008 [354]	DNA methylation	Vimentin methylation	86	82
Oberwalder/2008 [355]	DNA methylation	SFRP2 methylation	adenomas: 46	adenomas: 100
Glockner/2009 [356]	DNA methylation	tissue factor pathway inhibitor 2 (TFPI2) methylation	I–III stage of CRC: 76–89	I–III stage of CRC: 79–93
Melotte/2009 [357]	DNA methylation	NDRG4 methylation	61	93
Hellebrekers/2009 [358]	DNA methylation	GATA4/5 methylation	training set: 71 validation set: 51	training set: 84 validation set: 93
Ausch/2009 [359]	DNA methylation	*ITGA4* integrin, alpha 4 (antigen CD49D, alpha 4 subunit of VLA-4 receptor) methylation	adenomas: 69	adenomas: 79
Nagasaka/2009 [360]	DNA methylation	SFRP2 methylation RASSF2 methylation	CRC: 86 adenomas: 41 CRC: 45.3 adenomas: 12.6	94.7
Chang/2010 [361]	DNA methylation	ITGA4, SFRP2 methylation	CRC: 70 adenomas:72	panel: 96.8
Zhang/2011 [362]	DNA methylation	Vimentin, oncostatin M receptor (OMSR) and tissue factor pathway inhibitor 2 (TFPI2) methylation	CRC: 86.7 adenomas:76.5	86.7
Bosch/2012 [363]	DNA methylation	Phosphatase and Actin Regulator 3 (PHACTR3) methylation	training set: 55 CRC: 66 and adenomas (validation set): 32	training set: 95 validation set: 100
Ahlquist/2012 [132]	DNA methylation	BMP3, NDRG4, vimentin, TFPI2 methylation; mutant KRAS	adenomas: 82 CRC: 87	
Kisiel/2013 [364]	DNA methylation	BMP3 and NDRG4 methylation	CRC: 100 high grade dysplasia: 100 low grade dysplasia: 67	89
Amiot/2014 [365]	DNA methylation	Wif1, ALX4, vimentin methylation	Wif1:19 ALX4:11 vimentin:33	Wif1 and ALX4: 99 vimentin: 100
Imperiale/2014 [334]	DNA mutation, DNA methylation, DNA amount and protein testing	*K-ras* mutation, *BMP3* and *NDRG4* promoter methylation, DNA measurement by *β-* *actin* assessment and test for haemoglobin (FIT)	92.3	86.6
Zhang/2014 [366]	DNA methylation	SFRP2 methylation	CRC: 56.3 adenomas: 51.4	100
Wu/2014 [367]	DNA methylation	miR-34a methylation miR-34b/c methylation	76.8 95	93.6 100
Xiao/2014 [368]	methylation-sensitive high-resolution melting (MS-HRM)	*SNCA* and *FNB1* genes *Vimentin (VIM)* and *SFRP2* genes	84.3 92.5	93.30 91.2
Teixeira/2015 [369]	human DNA	total human DNA	66	89.8
Li/2015 [370]	DNA methylation	hypermethylated SNCA and FBN1	84.3	93.3
Park/2017 [371]	DNA methylation	methylated *SFRP2*, *TFPI2*, *NDRG4*, *BMP3* promoters	CRC: 94.3 adenomas: 72.2	55
Mojtabanezhad/2018 [347]	DNA methylation	SFRP1 and SFRP2 methylation	CRC: 56.5 adenomas:32.6	93.2
Sun/2019 [372]	DNA methylation	methylation of SDC2 and SFRP2, KRAS mutations and hemoglobin	91.4	86.1
Liu/2019 [373]	DNA methylation	methylation levels of SFRP2, SFRP1, TFPI2, BMP3, NDRG4, SPG20, and BMP3 plus NDRG4 genes	70	80
Bosch/2019 [337]	DNA methylation	*K-ras* mutation, *BMP3* and *NDRG4* promoter methylation, and hemogloblin	precancerous lesions: 46	89
Chen/2019 [374]	DNA methylation	SEPT9, NDRG4, SDC2	CRC: 90 adenomas: 78	
Liu/2020 [375]	DNA methylation	COL4A1, COL4A2, TLX2, and ITGA4	82.5–92.5	88.0–96.4
Jin/2020 [376]	DNA methylation	NDRG4, SDC2	81.82	93.75
Zhao/2020 [377]	DNA methylation	SEPT9, SDC2	92.3	93.2

**Table 8 ijms-23-00852-t008:** According to Loktionov in modification noninvasive stool miRNA biomarker for CRC detection [1].

Author/Year	Marker and Detection Method	Sensitivity [%]	Specificity [%]
Koga/2010 [397]	miR-17-92 cluster, upregulated miRNA panel: miR-17-92 cluster (miR-17, miR-18a, miR-19a, miR-19b, miR-20a, and miR-92a), miR-21, and miR-135, upregulated	69 74	81 79
Kalimutho/2011 [398]	miR-144 *	74	87
Wu/2012 [75]	miR-92a, upregulated miR-21, upregulated	71 (CRC) 56 (A) 55	73 73
Ahmed/2013 [399]	miR-7, miR-17, miR-20a, miR-21, miR-92a, miR-96, miR-106a, miR-134, miR-183, miR-196a, miR-199a-3p, miR214, miR-9, miR-29b, miR-127-5p, miR-138, miR-143, miR-146a, miR-222 and miR-938- findings: upregulated	N/A *	N/A *
Koga/2013 [18]	miRNA -106a upregulated and iFOBT	34.2	97.2
Wu/2014 [74]	miR-135b, upregulated	78 (CRC), 73 (AA) 62 (A)	68
Yau/2014 [400]	miR-221,upregulated miR-18a, upregulated	62 61	74 69
Phua/2014 [401]	miR-223 miR-451 both upregulated	77 88	96 100
Slaby/2016 [91]	miR-18a, upregulated miR-20a, upregulated miR-21, upregulated miR-92a, upregulated miR-106a, upregulated miR-135b, upregulated miR-144 *, upregulated miR-221, upregulated miR-223 and miR-92a, both upregulated panel miR-17-93 cluster and miR-135b, all upregulated	61 55 56 72 34 78 74 62 97 74	69 82 73 73 97 68 87 74 75 79
Chang/2016 [102]	miR-223, upregulated miR-92a, upregulated	73 71	82 79
Zhu/2016 [402]	miR-29a, miR-223, miR-224, **finding**: reduced expression in CRC stools	N/A	N/A
Yau/2016 [403]	miR-20a, upregulated	55	82
Wu/2017 [404]	miRNA panel: miR-144-5p, miR- 451a miR-20b- 5p, all upregulated	66	95
Bastaminejad/2017 [405]	miR-21, upregulated	86	81
Choi/2019 [406]	miR-21,upregulated miR-92a, upregulated miR-144 *, upregulated miR-17-3p, upregulated	79 89 78 67	48 51 66 70
Li/2020 [407]	miR-135b-5p, upregulated	96	74
Duran-Sanchon/2020 [408]	miR-421 and miR-27a-3p, both upregulated, finding: AUC -0.93 (for CRC)	N/A	N/A

* N/A—not applicable.

## Data Availability

Not applicable.

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
