# Peer review of "Novel Diagnostic Biomarkers in Colorectal Cancer"

_ijms, 2022, doi:10.3390/ijms23020852_

Round 1
Reviewer 1 Report
Minor points;
1)2.3. CIRCULATING TUMOR DNA (ctDNA)、line 1 and 2.4. CIRCULATING MICRORNA (c miRNA), line2. XX century is not a common expression.
2) 4.2. FECAL IMMUNOCHEMICAL TEST (FIT). FIT or FIT of two consecutive shows better result for finding CRC (Eur J Cancer Prev 2020 Mar;29(2):127-133, Cancer 2016 Jan 1;122(1):71-7).
Author Response
Manuscript ID: ijms-1518725 Type of manuscript: Review Title: Novel Diagnostic Biomarkers in Colorectal Cancer. Authors: Aneta Lidia Zygulska *, Piotr PierzchalskiReviewer #1:
1. 2.3. CIRCULATING TUMOR DNA (ctDNA)、line 1 and 2.4. CIRCULATING MICRORNA (c miRNA), line2. XX century is not a common expression.
We thank the Reviewer for the suggestion.
We have corrected and refilled first sentence in 2.3.Circulating tumor DNA (ctDNA)and third sentence in 2.4. Circulating microRNA (cmiRNA).
2) 4.2. FECAL IMMUNOCHEMICAL TEST (FIT). FIT or FIT of two consecutive shows better result for finding CRC (Eur J Cancer Prev 2020 Mar;29(2):127-133, Cancer 2016 Jan 1;122(1):71-7).
We are grateful the Reviewer for the constructive suggestion. We refilled 4.1. Guaiac-based fecal occult blood testing (gFOBT) and 4.2. Fecal immunochemical test (FIT) section. (Due to the fact, that the paper from Cancer was concerning the iFOBT, we put it in 4.1. section).
We added two suggested items of references in References section.

Reviewer 2 Report
Colorectal cancer (CRC) is one of the most prevalent and incident cancers worldwide with a significant morbidity and mortality. Less than a half of cases are diagnosed when the cancer is locally advanced. As we all know CRC is a heterogenous disease associated with a number of genetic and somatic mutation. It is very important to established new molecular noninvasive tests based on the detection of CRC alterations. Identifications of such molecular markers as DNA, RNA and proteins would improve survival rates and contribute to a personalized medicine.
This current article discussed the recent advances in novel diagnostic biomarkers for tumor tissue, blood and stool samples in CRC patients. The title and abstract of the article are representative in the context of the study topic and represent a point of reference in current research. The subject of the study is topical with real interest for the future. The introduction of the study is well outlined and the materials and methods used have a good reproducibility.
This current article presents in an objective way new screening methods in CRC. Blood biomarkers are detected in blood-based protein quantification tests or immunohistochemistry. Locally advanced malignant lesions enhance the level of circulating nucleic acids by up to 15-fold, where the concentration in patients with metastatic cancers can reach up to 500 ng/mL.Liquid biopsy can be used as a screening method to detect early-stage CRC and minimal residual disease after surgery or identify the molecular profile of CRC, the risk of recurrence, therapeutic targets, mechanism of drug resistance and influence the change of therapy. Circulating tumor cells (CTC) detection is a highly sensitive assay combining immunomagnetic enrichment with multiparameter flow cytometric and immunocytochemical analysis. Circulating tumor DNA (ctDNA) has been widelyevaluated as a novel biomarker for liquid biopsy in colorectal cancer diagnosis, prognosis and monitoring of response to treatment. Sensitivity of ctDNA for revealing of clinically relevant KRAS gene mutations was 87.2 % and its specificity 99.2%. An important pillar in non-invasive CRC screening is the development of protocols with a global impact with a certain economic and diagnostic feasibility. This present article is written in a clear and concise manner and highlighted the feasibility of some novel screening or diagnostic biomarker in CRC with precision and clarity. It is important to support the need of further studies in order to reach a definitive conclusion.
The article presents originality, with an optimal literary exposition, representing a topic of real interest for the future with objective results at the research level. The article represents a launching platform in its field and from the point of view of the characteristics it is included for publication
Author Response
Manuscript ID: ijms-1518725 Type of manuscript: Review Title: Novel Diagnostic Biomarkers in Colorectal Cancer. Authors: Aneta Lidia Zygulska *, Piotr Pierzchalski
Reviewer #2:
This current article presents in an objective way new screening methods in CRC (…)
This present article is written in a clear and concise manner and highlighted the feasibility of some novel screening or diagnostic biomarker in CRC with precision and clarity (…)
The article presents originality, with an optimal literary exposition, representing a topic of real interest for the future with objective results at the research level. The article represents a launching platform in its field and from the point of view of the characteristics it is included for publication.
We are very grateful the Reviewer for the positive review.